# Biological data assimilation for parameter estimation of a phytoplankton functional type model for the western North Pacific

*Yasuhiro Hoshiba[1,2], Takafumi Hirata[1], Masahito Shigemitsu[3], Hideyuki Nakano[4], Taketo Hashioka[3], Yoshio Masuda[1], Yasuhiro Yamanaka[1]

[1]Faculty of Environmental Earth Science, Hokkaido University, Japan

[2]Atmosphere and Ocean Research Institute, The University of Tokyo, Japan

[3]Japan Agency for Marine-Earth Science and Technology

[4]Meteorological Research Institute, Japan Meteorological Agency

*Correspondence to:* Yasuhiro Hoshiba (hoshi-y@aori.u-tokyo.ac.jp)

**Abstract.** Ecosystem models are used to understand ecosystem dynamics and ocean biogeochemical cycles and require optimum physiological parameters to best represent biological behaviours. These physiological parameters are often tuned up empirically, while ecosystem models have evolved to increase the number of physiological parameters. We developed a three-dimensional (3D) lower trophic level marine ecosystem model known as the Nitrogen, Silicon and Iron regulated Marine Ecosystem Model (NSI-MEM) and employed biological data assimilation using a micro-genetic algorithm to estimate 23 physiological parameters for two phytoplankton functional types in the western North Pacific. The estimation of the parameters was based on a one-dimensional simulation that referenced satellite data for constraining the physiological parameters. The 3-D NSI-MEM optimised by the data assimilation improved the timing of a modelled plankton bloom in the subarctic and subtropical regions compared to the model without data assimilation. Furthermore, the model was able to improve not only surface concentrations of phytoplankton but also their subsurface maximum concentrations. Our results showed that surface data assimilation of physiological parameters from two contrasting observatory stations benefits the representation of vertical plankton distribution in the western North Pacific.

## 1 Introduction

The Western North Pacific (WNP) region is a high-nutrient, low-chlorophyll (HNLC) region where biological productivity is lower than expected for the prevailing surface macronutrient conditions. There are both Western Subarctic Gyre and Subtropical Gyre comprising the Oyashio and the Kuroshio, respectively (Fig. 1 (a)). Between the gyres (i.e., the Kuroshio–Oyashio transition region), horizontal gradients of temperature and phytoplankton concentration in the surface water are generally large due to meanders in the Kuroshio extension jet and mesoscale eddy activity (Qiu and Chen, 2010; Itoh et al., 2015). The relatively low productivity in the HNLC region is due to low dissolved iron concentrations (e.g., Tsuda et al., 2003), because iron is one of the essential micronutrients for many phytoplankton species. The source of iron for the WNP region is not only from air-born dust but also from iron transported in the intermediate water from the Sea of Okhotsk to the Oyashio region (Nishioka et al., 2011). Since the WNP region exhibits many complex physical and biogeochemical characteristics as referred to above, it is difficult even for state-of-the-art eddy-resolving models to reproduce them.

Processes of growth, decay, and interaction by plankton are critical to understanding the oceanic biogeochemical cycles and the lower trophic level (LTL) marine ecosystems. There are many LTL marine ecosystem models ranging from simple nutrient, phytoplankton and zooplankton models to more complicated models including carbon-, oxygen-, silicate-, iron-cycles and so forth (e.g., Fasham et al., 1990; Edwards and Brindley, 1996; Lancelot et al., 2000; Yamanaka et al., 2004; Blauw et al., 2009). Coupling LTL marine ecosystem models to ocean general circulation models (OGCMs) and earth system models enables three-dimensional (3D) quantitative descriptions of the ecosystem and its temporally fine variability (e.g., Aumont and Bopp, 2006; Follows et al., 2007; Buitenhuis et al., 2010; Sumata et al., 2010; Hoshiba and Yamanaka, 2016).

Physiological parameters are usually fixed in the models on the basis of local estimations and applied homogeneously to a basin-scaled ocean, although the values of physiological parameters should depend on the environments of regions. Moreover, physiological parameters have been often tuned up empirically and arbitrarily. The fact that the number of parameters increases with prognostic and diagnostic variables makes it more difficult to tune them. In order to reproduce observed data such as spatial distribution of phytoplankton biomass and timing of a plankton bloom, it is required to reasonably estimate the physiological parameters.

In previous studies using LTL marine ecosystem models, various approaches for data assimilation were introduced as methods of estimating optimal physiological parameters (e.g., Kuroda and Kishi, 2004; Fiechter et al., 2013; Toyoda et al., 2013; Xiao and Friedrichs, 2014). On the other hand, Shigemitsu et al. (2012) applied a unique assimilative approach to an LTL marine ecosystem model, using a micro-genetic algorithm (μ-GA) (Krishnakumar, 1990). For the western subarctic Pacific, they showed that the μ-GA worked well in the one-dimensional (1D) nitrogen-, silicon- and iron regulated marine ecosystem model (NSI-MEM: Fig. 2), that was based on NEMURO (North pacific Ecosystem Model for Understanding Regional Oceanography: Kishi et al., 2007) but differed in the following points: (1) the introduction of an iron cycle, including dissolved and particulate iron, whereby the dissolved iron explicitly regulates phytoplankton-photosynthesis; (2) adoption of physiologically more consistent optimal nutrient-uptake (OU) kinetics (Smith et al., 2009) instead of the Michaelis–Menten

equation (Michaelis et al., 2011) and (3) the division of detritus into two types of small and large sizes that exhibit different
sinking rates.
Our objective is to improve simulation of the LTL ecosystem in the WNP region by further introducing: (1) a physical field
from an eddy-resolving OGCM with a horizontal resolution of 0.1° and (2) an assimilated physiological parameter estimation
for two different phytoplankton groups. The details of the model and μ-GA settings are described in Section 2. We compare
the simulation results with/without the parameter optimisation to observed data and confirm the effects of changing parameters
in Section 3. We mainly focused on the seasonal variations of phytoplankton in the pelagic region. Finally, the results are
summarized in Section 4.
**2 Model and data description**
**2.1 3D NSI-MEM**
We used the marine ecosystem model, NSI-MEM that includes two phytoplankton functional types (PFTs), namely non-
diatom small phytoplankton (PS) and large phytoplankton representing diatoms (PL) (Fig. 2). In order to run the NSI-MEM
in three-dimensional space, we used a physical field obtained from the Meteorological Research Institute Multivariate Ocean
Variational Estimation for the WNP region (MOVE-WNP) (Usui et al., 2006). The MOVE-WNP system is composed of the
OGCM (the Meteorological Research Institute community ocean model) and a multivariate 3D variational (3DVAR) analysis
scheme. The 3DVAR method adds some increments to only the temperature and salinity fields. The increments are derived so
as to minimize the misfits between the model and observations of temperature, salinity and sea surface dynamic height (Fujii
and Kamachi, 2003). The dynamical fields such as flow speed and sea surface height are not directly modified by the 3DVER
(i.e., the physical field holds water mass conservation, which is necessary to run the ecosystem model with a consistent manner).
The model domain extends from 15° N to 65° N and 117° E to 160° W in the WNP region, with a grid spacing of 1/10° ×
1/10° around Japan and 1/6° to the north of 50° N and to the east of 160° E (Fig. 1 (a)). There are 54 vertical levels with layer
thicknesses increasing from 1 m at the surface to 600 m at the bottom. The model was forced by factors including surface wind,
heat flux, and freshwater flux. The details of the surface forcing are presented by Tsujino et al. (2011). Shortwave radiation
input and dust flux were the same as those of a global climate model (Model for Interdisciplinary Research on Climate,
MIROC; Watanabe et al., 2011). A part of the dust flux (3.5 %; Shigemitsu et al., 2012) was regarded as the iron dust, and
1 % of the iron dust was assumed to dissolve into the sea surface (Parekh et al., 2004). The other iron dust was transported to
the lower layers and dissolved, which was the same process as Shigemitsu et al. (2012). River run-off as a freshwater supply
was from CORE ver. 2 forcing (Large and Yeager, 2009), in which the river source had the nitrate concentration value of 29
μmol/l (Conha et al., 2007) and the silicate concentration value of 102 μmol/l adjusted in the range between Si/N = 0.2 to 4.3
(Jickells, 1998). Nitrate and silicate sources were only rivers, and iron supply was only from the dust in the model setting. In
order to buffer artificial high concentrations near the side edge of the model domain, nutrients near the southern and eastern
boundary of the model domain were only restored for 43 minutes to 3.6 hours to the values provided by the Meteorological
Research Institute Community Ocean Model (MEM-MRI.COM) participating in MARine Ecosystem Model Intercomparison
Project (https://pft.ees.hokudai.ac.jp/maremip/data/MAREMIPh_var_list.html). The physical field used in our ecosystem
model had already been confirmed to reproduce realistic salinity, velocity and temperature fields in a previous study (Usui et
al., 2006). Using a physical one-day averaged field, we ran the NSI-MEM to simulate the years between 1985 and 1998.
We divided the model domain into two provinces (green and yellow regions in Fig. 1 (b)) using the following province map
instead of maps divided by latitude-longitude lines as in previous studies (e.g., Longhurst, 1995; Toyoda *et al.*, 2013). The
province map is based on the dominant phytoplankton species and nutrient limitations (Hashioka et al., in preparation) and
sets different ecosystem parameters (see details in Section 2.3) for each province (hereafter, 'Parameter-optimised case: OPT';
Table 1). For each province, the respective parameters estimated by the µ-GA and the 1D NSI-MEM were employed to
those in the 3D NSI-MEM. A large gap in a horizontal-distribution of phytoplankton can appear on the
boundary of the two provinces in Fig. 1 (b), due to a gap in the different parameter sets at the boundary. In
order to smooth the gap in parameter values at the boundary between the two provinces in Fig. 1 (b), the parameters were
varied as a function of the sea surface temperature (SST) annually averaged for 1998 (Fig. 1 (c)) for our 'SST-dependent case:
SST-OPT' (Table 1). While phytoplankton fluctuate with not only SST but also other surrounding conditions such as nutrient
abundance in the real ocean (Smith and Yamanaka, 2007; Smith et al., 2009), we chose SST because µ-GA optimization is
conducted for physiological parameters of both phytoplankton and zooplankton (Table 2) and the SST directly affects
physiology of both of them whereas nutrients and light were essentially related to phytoplankton. The parameters were
interpolated/extrapolated according to the following equation:

$$P(x) = P_{St.S1} + \left(P_{St.KNOT} - P_{St.S1}\right) \times \frac{SST(x) - SST_{St.S1}}{SST_{St.KNOT} - SST_{St.S1}}, \qquad (1)$$

where $P(x)$, $P_{St.\,S1}$ and $P_{St.\,KNOT}$ are ecosystem parameters for a point ($x$), St. S1 and St. KNOT, respectively. St. KNOT and
St. S1 are typical observational points in the subarctic and subtropical regions (green- and yellow-coloured areas in Fig. 1 (b),
respectively). We also conducted model experiments with the parameters similar to Shigemitsu et al. (2012) for the whole
domain (hereafter 'Control case: CTRL', Table 1). The parameters of all the 3D experimental cases, shown in Table 1, were
not changed either vertically or temporally. In the parameter-optimised and SST-dependent cases, the parameters were the
same as the Control case from 1st January 1985 to 31st December 1996. During the next one year (1997), the simulations were
spun-up with the optimised or SST-dependent parameters. Then, simulation results on 1st Jan. 1998 were used as initial
conditions for the 1998-year simulations. The parameter values used in the control case were not changed during the 1985-to-
1998 period. The simulation results for the last year (i.e., 1998) were analysed and compared to observational data of 1998.
**2.2 Satellite and in situ data**
Global satellite data for 1998 for phytoplankton (i.e., chlorophyll a) were obtained from the Ocean Colour Climate Change
Initiative, European Space Agency, available online at http://www.esa-oceancolour-cci.org/, which utilised the data archives
of ESAs MERIS/ENVISAT and NASAs SeaWiFS/SeaStar, Aqua/MODIS. The global satellite data which have the horizontal
resolution of 0.042° were linearly interpolated to the grid (size 1/10° and 1/6°) in the model domain (Fig. 1 (a)), and the
nitrogen-converted concentrations of both PL and PS were estimated based on a satellite PFT algorithm (Hirata et al., 2011).
The μ-GA cost function was defined from the 1998 monthly averaged PL and PS concentrations. The satellite data of daily
temporal resolution were not useful due to many regions of missing value. Therefore, we discuss the results of the monthly
scale in the present study.
Satellite data of the 1998 mean SST (horizontal grids of 0.088°) from the AVHRR Pathfinder Project
(http://www.nodc.noaa.gov/SatelliteData/pathfinder4km/) were also used to conduct our SST-dependent case study using the
same interpolation procedure as the above. The data were linearly interpolated between satellite- and model grids, which could
introduce some uncertainty to the satellite data. In addition, the use of the global chlorophyll data in the regional study for the
WNP region could be another error source of the observational data: the previous study (Gregg and Casey, 2004) showed that
the regional Root Mean Square log % errors of the satellite data ranged from 24.7 to 31.6 in the North Pacific.
To validate the vertical distribution of the model results, we utilised in situ data of phytoplankton and nutrients in 1998 along
165° E section taken from World Ocean Database 2013 (https://www.nodc.noaa.gov/OC5/WOD13/), and at St. KNOT (44° N,
155° E) obtained from the website (http://www.mirc.jha.or.jp/CREST/KNOT/) (Tsurushima et al., 2002).
**2.3 1D NSI-MEM process**
The 1D NSI-MEM used in Shigemitsu et al. (2012) was employed as an emulator to determine the optimal set of ecosystem
parameters at St. KNOT (44° N, 155° E) and S1 (30° N, 145° E), respectively. We modified the 1D NSI-MEM of Shigemitsu
et al. (2012) by increasing the number of vertical layers to 54 and introducing the vertical advection of the 3D simulation.
Twenty-three of 107 physiological parameters in the NSI-MEM were selected, as shown in Table 2, which were responsible
for PL and PS biomass relevant to the photosynthesis and the grazing of zooplanktons. In the previous study, Yoshie et al.
(2007) also suggested that some parameters in the 23 parameters were relatively influential on PS and PL, more than the other
physiological parameters such as those for sinking process of particulate matters (PON, OPAL in Fig. 2). The other parameters
of the NSI-MEM were the same as those in the Control case. The initial (1st January 1998) and boundary conditions during the
integration period were applied from those in the 3D model.
**2.4 μ-GA implementation**
The μ-GA procedure requires a cost function. To define the cost function (Eq. (2)), satellite PFT data were used as reference
values for the μ-GA because satellite data have higher temporal and spatial resolution than in situ data. The μ-GA procedure
works in such a way that a parameter set of the lowest cost is retained, and then a new parameter set is determined by crossover
and mutation methods using the retained set. An optimised parameter set is finally provided by repeating the process multiple
times.
Running the 1D NSI-MEM with the μ-GA, the 23 optimal parameters were obtained through the following process:
*Step 0*    Define a range of parameter values (Table 2) based on previous studies (e.g., Jiang et al., 2003; Fujii et al., 2005;
Yoshie et al., 2007) and prepare 23 model runs being the same number of estimated parameters before running the µ-GA.
*Step 1*   Generate 23 initial random parameter sets using the µ-GA.
*Step 2*   Evaluate the 23 model runs with the different parameter sets using the following cost function:

$$Cost = \sum_{i}^{I} \frac{1}{N_i} \sum_{j}^{Ni} \frac{1}{\sigma^2{}_i}(m_{ij} - d_{ij})^2 , \qquad (2)$$


where $m_i$ is the modelled monthly mean of phytoplankton type $i$ ($i = 1$ for PL and 2 for PS) and $d_i$ is the monthly satellite data
of type $i$. The index $j$ denotes the number of months ($N_i$) for which satellite data of type $i$ exists. The assigned weights for PL
and PS were the same low value ($\sigma_{PL} = 0.1$ µmol/l and $\sigma_{PS} = 0.1$ µmol/l) as some weights used in Shigemitsu et al. (2012).
*Step 3*   Determine the best parameter set and carry it forward to the next model run (or the next 'generation') (elitist strategy).
*Step 4*   Choose the remaining 22 sets for re-determination of the best parameter sets (or 'reproduction') based on a
deterministic tournament selection strategy (the best parameter set that gave the highest model performance in Step 3 also
competes for its copy in the reproduction). In the tournament selection strategy, the parameter sets are grouped randomly and
adjacent pairs are made to compete. Apply crossover to the winning pairs and generate new parameter sets for the final 22
parameter sets. Two copies of the same set mating for the next generation should be avoided.
*Step 5*   If the difference between the maximum and minimum cost function values of the model runs becomes smaller than
a threshold value, renew all the parameter sets randomly except for the best-performed set for efficiently escaping from a local
solution; the cost function may have local minimums.
*Step 6*   Repeat the procedure from Step 2 to Step 5 until the best parameter set is well converged within 2,000 generations
(times) in the present study.
The 1D NSI-MEM was used as an emulator to determine ecosystem parameters through the process described above, and the
parameter sets assimilated by the 1D model with the µ-GA at St. KNOT and St. S1 were applied to the 3D simulations which
were conducted as the Parameter-optimised case and the SST-dependent case in Table 1.
**3. Results and discussion**
**3.1 1D model**
The 1D NSI-MEM was employed to determine ecosystem parameters for the 3D-model simulation. The 1D simulation results
(Fig. 3) of Parameter-optimised case (blue dashed lines) are clearly closer to satellite data (solid lines) than those of Control
case (orange dashed lines). The cost-function values estimated by the 1D simulations in the Parameter-optimised case (OPT),
1.61 and 0.17 at KNOT and S1, are also about 8 and 6 times smaller than those in the Control case (CTRL), 13.55 and 1.11,
respectively (not shown).
The total biomass (PL+PS) at St. KNOT in the subarctic region is larger than that at St. S1 in the subtropical region. The PS
biomass (Fig. 3 (a), (c)) is larger than the PL biomass (Fig. 3 (b), (d)) at both St. KNOT and St. S1. As for the relative ratio of
PL to the total biomass, the relative ratio at St. KNOT is larger than that at St. S1. These results are consistent with the general

understanding that biomass in the subarctic region is larger than that in the subtropical region, and that the ratio of PL to the total biomass in the subarctic region is also larger than that in the subtropical region.

Seasonal variations in the OPT for the two stations simulated with the satellite data assimilation are also improved drastically in comparison to the CTRL. The seasonal variations of PS and PL at St. KNOT (Fig. 3 (a), (b)) in the OPT have relatively high concentrations with a winter peak of 630 $\mu$molN/m$^3$ and 130 $\mu$molN/m$^3$, respectively. In the CTRL of PS, however, there is a spring (May) peak of 180 $\mu$molN/m$^3$, and the PL concentration remains low through the year. At St. S1, the PS seasonal variations tend towards high-concentration in winter and low concentration from summer to autumn in the OPT, while the PS concentration, in the CTRL, in summer to autumn is higher than that in winter. The PL concentrations of the two model cases are almost zero, and that of the satellite is also remarkably small ($< 21.5$ $\mu$molN/m$^3$). The parameter-optimisation process by 1D model works well in terms of the seasonal variations of surface phytoplankton.

**3.2 3D model**

The parameter set estimated by the 1D model at St. KNOT and St. S1 were applied to the 3D simulation (Fig. 4). The seasonal features in the 3D simulation are generally similar to those seen in the 1D simulation (i.e., relatively small seasonal variations of PS biomass in the subarctic region and a relatively high winter biomass in the OPT, than the CTRL). At St. KNOT, for instance, there is the smaller difference between the high (575 $\mu$molN/m$^3$ in January) and low (398 $\mu$molN/m$^3$ in October) concentrations in the OPT than the high (568 $\mu$molN/m$^3$ in July) and low (59 $\mu$molN/m$^3$ in January) in the CTRL. The PL biomass features are also similar to those of the PS biomass mentioned above, except that the PL biomass is lower in the subtropical region in the OPT than in the CTRL. Seasonal peaks of PS and PL biomass also have the same features as those in the 1D simulations (i.e., the PS bloom in the OPT occurs from winter to spring (Fig. 4 (c), (g)), but that in the CTRL occurs in summer (Fig. 4 (b)). The SST-dependent (SST-OPT) results are discussed later in Section 3.5.

Higher phytoplankton concentrations ($> 1000$ $\mu$molN/m$^3$) were found in coastal areas throughout the year in the satellite data. The model could not simulate these high concentrations in the coastal areas. This may be due to the inaccuracy of the satellite data resulting from the high concentrations of dissolved organic material and inorganic suspended matter (e.g., sand, silt, and clay), and/or due to the uncertainty in the model introduced by unaccounted coastal dynamics such as small-scale mixing processes (e.g., estuary circulation, tidal mixing and wave by local wind forcing). Any nutrient flux from the seabed was not considered in this study, which also may induce the low-biased phytoplankton biomass close to the coast.    Hereafter, we focus on phytoplankton seasonal fluctuation in the pelagic and open ocean in this study.

Lagged (within ±2 months) correlation coefficients were calculated for the monthly time series of the surface phytoplankton concentration between the simulations and satellite data in each grid (Fig. 5 (a), (c), (e)). Although there are some regions where the correlation values are out of the range in the 95 % significance level (Fig. 5 (b), (d), (f)) due to the small numbers of monthly mean data, the correlation maps of CTRL, OPT, and SST-OPT can be relatively comparable each other, because of the same sample numbers of the simulations in each grid. Spatial distributions of the correlation show that the larger coefficient-value region (r > 0.7) of the OPT (Fig. 5 (c)) in 25° N -45° N becomes extended than that of the CTRL (Fig. 5 (a)) by 71 %,

though the mean value of the OPT in the north part of 50° N (r = 0.18) is smaller than that in the CTRL (r = 0.66). The result
is similar in the SST-OPT (Fig. 5 (e)). Our parameter estimation significantly improves the simulation result of the horizontal
distribution of phytoplankton in the lower latitude (< 45° N), but not in the region (> 50° N) closer to the coasts.
Fig. 6 (a)-(c) shows vertical distributions of total phytoplankton along the 165° E transect. The parameter optimisation
improves the distributions in that the phytoplankton maximum in the subsurface more deepens than that of CTRL (Fig. 6 (b),
(c)). Parameter-optimised total biomass through the vertical section above 200 m is also closer to the observed data than the
CTRL. It is an interesting result because the vertical distribution is improved due to the data-assimilation process using only
surface satellite data. The detailed reason is discussed in Section 3.4. In the nutrients distribution along the 165° E (Fig. 6 (d)
to (i)), the concentrations of OPT (Fig. 6 (f), (i)) are lower than those of CTRL (Fig. 6 (e), (h)). The mean values along the
transect of nitrate and silicate are 0.011 molN/m$^3$ and 0.025 molSi/m$^3$, respectively, in the OPT, 0.014 molN/m$^3$ and 0.034
molSi/m$^3$ in the CTRL, and 0.012 molN/m$^3$ and 0.022 molSi/m$^3$ in the observation (Fig. 6 (d), (g)). OPT than CTRL is better
consistent with the observation, though the nitrate observed value is higher than the simulations in the surface (< 80 m) and
subarctic (> 42° N) region. While nitrate is not the limiting nutrient compared with iron and silicate for phytoplankton's
photosynthesis in the subarctic region (the detail is also mentioned in Section 3.4), the data-assimilation process improves even
the nutrient field in addition to the phytoplankton field.
As for the temperature and salinity along the vertical section (Fig. 7), the physical field used by the model simulations is well
reconstructed in terms of mixed layer depth and transition from the subarctic and the subtropical regions. Judging from the
temperature and salinity distributions in the subarctic region (> 42° N), the water columns are well mixed vertically both in
the observation and the simulation, and intensely stratified in the subtropical region (< 36° N). There is the transition region
(36° N -40° N) of temperature between the subtropical and the subarctic.

**3.3 Amplitude and phase of seasonal variation of phytoplankton**

The model performances were significantly improved in terms of spatial distributions of phytoplankton biomass, as a result
of the parameters optimised in Section 3.2. Also at the specific stations on the St. KNOT and St. S1 where the parameters were
estimated by the 1D simulations, seasonal variations in total phytoplankton concentrations in the OPT are generally better
reproduced to those in the satellite data than those in the CTRL (Fig. 8). At St. KNOT (Fig. 8 (a)), the phytoplankton bloom
in the OPT occurs in winter, and the phytoplankton bloom in the CTRL occurs in summer in an antiphase to that of the satellite.
At St. S1 (Fig. 8 (b)), OPT case reasonably captures the timing of the phytoplankton bloom by the satellite, although the
amplitude is slightly overestimated. The seasonal variations of the PS and PL concentrations are similar to those of the total
phytoplankton (not shown) in both cases.
Figure 9 shows comparisons of the amplitude and the phase of seasonal variations between three model cases (CTRL, OPT,
and SST-OPT) and the satellite data. The radius shows the amplitude of seasonal variation for each of the modelled cases
relative to the satellite data, and the angle from the x-axis shows the maximum concentration time lag for each of the model
cases (i.e., the point (1, 0) shown as 'True' is a match within 1 month and 30 degrees error range to the satellite data). At St.
KNOT, the OPT (blue solid vector) exhibits the phase closest to the satellite data among the three modelled cases. The ratios
of the amplitudes to the satellite data are as follows: 1.00 for the OPT (blue solid vector), 1.08 for the SST-OPT (yellow solid
vector) and 1.24 for the CTRL (orange solid vector). The timings of the maximum concentration are as follows: a two-month
delay for the OPT, a three-month delay for the SST-OPT and a six-month delay (anti-phase) for the CTRL. The timing of the
OPT at St. S1 (blue dotted vector) is improved, though its seasonal amplitude is not.
Optimisation of the physiological parameters by assimilating the satellite data at the two stations improves the seasonal
variations of the phytoplankton concentrations such as the timing of the maximum concentration and the seasonal amplitude
of the WNP region.
**3.4 Vertical distributions of phytoplankton and nutrients concentrations at St. KNOT**
The model-simulated vertical distributions of phytoplankton, nitrate and silicate concentrations at St. KNOT on $20^{th}$ July
1998 were compared with the observed ones on the same day (Fig. 10). The vertical distribution of phytoplankton (Fig. 10 (a))
by 3D simulations in the OPT (solid blue line) is closer to the in situ data (black line) as compared to the CTRL (solid orange
line): the maximum phytoplankton concentration for the OPT and the in situ data are located in the subsurface around a depth
of 50 m, while there is no subsurface maximum in the CTRL. The differences in the biomass between the OPT and CTRL
become especially larger in the subsurface layer (40 m to 80 m). Thus, better physiological parameterisation through the data
assimilation improves not only the surface concentration but also the important characteristics of vertical plankton distribution
such as the subsurface maximum. This is an interesting improvement because the physiological parameters are optimised using
only surface satellite data.
The vertical profile of phytoplankton obtained from the 3D simulation reproduces the observed ones better than the 1D
simulation, too (Fig. 10 (a)). In addition, the difference in 3D (solid lines) and 1D (dashed lines) is larger in the upper layer (<
80 m) than in the lower layer (> 100 m). Moreover, error bars and shade for the 3D simulations, which depict the maximum
and minimum values in ± 0.3° around the exact grid of St. KNOT, are also larger in the upper layer than the lower layer. We
assume that horizontal advection such as mesoscale eddies is in the O (100 km) radius scale and ⩾16 weeks' lifetime (e.g.,
Chelton et al., 2011) and can be detected within the ± 0.3° range in the physical field. These suggest that effects of horizontal
advection are important for the daily reconstruction of the profile in the upper layer as the effects are not included in the 1D
model.
In the NEMURO, the predecessor version of the NSI-MEM, the amplitude and timing of phytoplankton blooms are
predominantly controlled by the photosynthesis rate (i.e., bottom-up effect of nutrient dependence) rather than the grazing rate
(i.e., top-down effect of zooplanktons) (Hashioka et al., 2013). The former is determined by the limited growth rate which is a
limitation function of growth rate by either nitrogen ($NH_4$+$NO_3$), silicate ($Si(OH)_4$) or dissolved iron (FeD) (refer to Eq. (A15)
and Eq. (A23) in Shigemitsu et al., 2012). The smallest limited growth rate among the three nutrient groups (i.e., $NH_4$+$NO_3$,
$Si(OH)_4$ and FeD) is used to limit the rate of phytoplankton's photosynthesis. For PS and PL in the OPT and CTRL, the
dissolved-iron-limited growth rates (red lines in Fig. 11) dominate the photosynthesis, while the silicate-growth rate is the

second-largest limiting factor for PL (green lines in Fig. 11 (b)). The mean iron-growth rates increase remarkably below a depth of 50 m (e.g., 0.37 to 1.86 day$^{-1}$ and 0.48 to 2.47 day$^{-1}$ in PS and PL, respectively) because of the parameter optimisation of the potential maximum growth rate ($V_0$) and the affinity ($A_0$) as shown in Table 2. As a result, the uptake of dissolved iron seems to be accelerated, particularly in the subsurface layer, leading to an increase of the phytoplankton biomass (Fig. 10 (a)). The larger biomass of phytoplankton may also consume more nitrate and silicate nutrients resulting in a lower nitrate concentration above a depth of 140 m (Fig. 10 (b)) and silicate (Fig. 10 (c)) as compared to that in the CTRL. The vertical gradients of nitrate and silicate in the OPT are closer to the observed data than those in the CTRL. In the OPT, nitrate and silicate concentrations are less than the data in situ, both at the depth of around 50 m (0.010 molN/m$^3$ and 0.015 molSi/m$^3$ in the OPT; 0.015 molN/m$^3$ and 0.025 molSi/m$^3$ in the observation) and 200 m (0.031 molN/m$^3$ and 0.069 molSi/m$^3$; 0.038 molN/m$^3$ and 0.085 molSi/m$^3$, respectively), while those at the depth of around 50 m in the CTRL (0.017 molN/m$^3$ and 0.037 molSi/m$^3$) is higher than those in the observed data in which smaller vertical gradients of CTRL than the OPT are found. In the upper layer, the nutrients are adequately supplied to phytoplankton as a result of the parameter optimisation. As in the lower layer below the depth of 120 m, the nutrient concentrations seem to be also determined by physical processes in the ocean-basin scale, not only local biological processes.

The change of the dissolved-iron-limited growth rates by optimisation results in the lower concentration of dissolved iron in the subarctic area (Fig. 12) because of the greater consumption of FeD by the phytoplankton than in the CTRL. The result is so far consistent with the conception of an HNLC region in the North Pacific Ocean (Moore et al., 2013), in spite that our model does not include the iron source from the Sea of Okhotsk to the WNP region as another iron source (Nishioka et al., 2011). A further improvement is expected by adding such an iron supply into our model.

**3.5 Physiological parameter changes with ambient conditions**

The SST-OPT (i.e., smoothed changing parameters) was compared to the OPT (i.e., boundary-gap parameters). The horizontal distribution of the PS and PL concentrations in the SST-OPT are not significantly different from those in the OPT (Fig. 4) except in two regions—the western region of low latitude (15° N to 25° N and 120° E to 150° E during January and April in Fig. 4 (h)), and the region adjacent to the Kuroshio Extension (around 40° N during July to October in Fig. 4 (h)). The former exception is due to the extrapolation of parameters with high SST and the latter is due to smoothing of parameters between the St. KNOT and St. S1 stations. The simulated seasonal variations of phytoplankton concentration in the SST-OPT is slightly worse than those in the OPT at the two stations (Fig. 9). The ratios of the seasonal amplitudes at St. S1, for instance, were 2.33 for the OPT and 2.39 for the SST-OPT. The maximum concentrations for both cases are found in the same month (March) as that for the satellite data (they overlap each other on the no-lagged x-axis in Fig. 9). However, a smoothed set of parameters dependent on the SST prevents the artificial gap of the parameter value at the fixed boundary between the two provinces.

Physiological parameters represented in ecosystem models were optimised in reference to 1998 while they may change with time. In addition, they may change with the surrounding conditions in the real ocean (e.g., SST, nutrient abundance, and light intensity). Smith and Yamanaka (2007) and Smith et al. (2009) suggest the significance of photo-acclimation and nutrient

affinity acclimation. Phytoplankton cells change their traits (e.g., nutrient channel, enzyme) in response to ambient nutrient concentrations, and typically large (small) cells adapt to low (high) light and high (low) nutrient concentrations (Smith et al., 2015). In the NSI-MEM, the effect of nutrient-uptake responses by plankton acclimated to different ambient nutrient conditions is applied as an OU kinetic formulation, but the effect of photo-acclimation has not yet been introduced. Incorporation of temporal variation in the physiological parameters may be effective in precisely reproducing distributions and variations of phytoplankton. In other words, data assimilation through the physiological parameter change with environmental conditions might play part in a calibration of simplified formulations of LTL marine ecosystem models. However, four-dimensional changes of physiological parameters complicate scientific interpretation (Schartau et al., 2016), even though marine ecosystem models have been developed in order to simplify the marine ecosystem and facilitate scientific interpretation. The spatial parameter estimation was conducted in this study because we would like to also discuss the physiological effects of parameters changing in detail.

## 4 Conclusions

We extended an LTL marine ecosystem model, NSI-MEM, into a 3D coupled OGCM. We also used a data assimilation approach for two different PFTs in the WNP region: non-diatom PS and PL. Twenty-three ecosystem parameters in the NSI-MEM were estimated using a 1D emulator with a μ-GA parameter-optimisation procedure. By applying the optimised parameters to the 3D NSI-MEM, the model performances were improved in terms of the seasonal variations of phytoplankton biomass, including the timing of the plankton bloom in the surface layer, compared to those using prior parameter values (Control case). Notably, the vertical distribution of phytoplankton such as the subsurface maximum layer was also improved via the parameter optimisation, compared to that in the Control case. Thus, it was demonstrated that the 3D simulation performed better than the 1D simulation even to reproduce the vertical profile of phytoplankton.

Physiological parameters in this study were systematically determined by a μ-GA within the range of those used by numerical models in previous studies. While our parameter estimation improved the modelling skill of temporal and spatial variability of PL and PS in the WNP, the estimated parameter values themselves should also be confirmed with sufficient amount of data when they become available, in order to increase our confidence towards mechanistic and numerical understanding of the phytoplankton dynamics observed.

Acknowledgements

This study was supported by Core Research for Evolutional Science and Technology (CREST), Japan Science and Technology Agency, Grant Number JPMJCR11A5. The first author developed the 3D NSI-MEM and conducted simulations using this model at Hokkaido University and analysed the results supported by the Center for Earth Surface System Dynamics, Atmosphere and Ocean Research Institute, The University of Tokyo. The phytoplankton satellite data were gathered by the Ocean Colour Climate Change Initiative, ESA (European Space Agency). The SST-satellite data was provided by the National

Oceanic and Atmospheric Administration Pathfinder project in GHRSST (The Group for High Resolution Sea Surface

Temperature) and the US National Oceanographic Data Center. Data in situ used in this study were taken from World Ocean

Database 2013 and Ocean Time-series Program in western North Pacific.

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

Table

Table 1. List of experiments

| | Experiment name | Content of experiment |
|---|---|---|
| 1D model experiments | Control | Use the almost same parameters as those in Shigemitsu et al. (2012) |
| | Parameter-optimised | Optimise the parameters with μ-GA at St. KNOT and St. S1 |
| 3D model experiments | Control | The same as Control of 1-D model but applied to 3-D simulation |
| | Parameter-optimised | The same as Parameter-optimised of the 1-D model but applied to 3-D simulation for two provinces of Fig. 1 (b) |
| | SST-dependent | The same as Parameter-optimised of 3-D simulation with interpolated parameters at St. KNOT and St. S1 with SST, instead of parameters for two provinces |


Table 2. NSI-MEM physiological parameters estimated by the μ-GA. Max and Min values prescribe the upper and
lower bounds of the parameter variations used in the previous studies. St. KNOT and St. S1 indicate optimal
estimated values in the provinces of Fig. 1 (b) while Control values are not optimised parameter values,
and the values of Shigemitsu et al. (2012) are the parameters of the previous study.

| Parameter | Symbol | Min | KNOT | S1 | Control | Shigemitsu et al. (2012) | Max | Unit | Sources of Min and Max range |
|---|---|---|---|---|---|---|---|---|---|
| PS Potential maximum growth rate at 0℃ | $V_{0,PS}$ | 0.1 | 2.7 | 0.7 | 0.6 | 0.6 | 3.2 | /day | Shigemitsu et al. (2012) |
| PS Potential maximum affinity for NO₃ | $A_{0,NO3,PS}$ | 1 | 454 | 436 | 30 | 282 | 512 | l/molN・s | Shigemitsu et al. (2012) |
| PS Half satuation constant for NO₃ | $K_{NO3,PS}$ | 0.5 | 1.871 | 2.9194 | 1 | 1 | 3 | μmolN/l | Chai et al. (2002), Eslinger et al. (2000) |
| PS Half satuation constant for NH₄ | $K_{NH4,PS}$ | 0.05 | 0.1225 | 0.2582 | 0.1 | 0.1 | 1 | μmolN/l | Chai et al. (2002), Eslinger et al. (2000) |
| PS Half satuation constant for FeD | $K_{Fed,PS}$ | 0.035 | 0.1 | 0.0602 | 0.04 | 0.05 | 0.1 | nmol/l | Kudo et al. (2006), Price et al. (1994) |
| PS Temperature coefficient for photosynthetic rate | $k_{PS}$ | 0.0392 | 0.0693 | 0.065 | 0.0693 | 0.0693 | 0.0693 | /degC | Eslinger et al. (2000), Fujii et al. (2005) |

| Description | Symbol | | | | | | | Unit | Reference |
|---|---|---|---|---|---|---|---|---|---|
| PS Mortality rate at 0℃ | $M_{PS0}$ | 0.012075 | 0.012075 | 0.043212 | 0.0585 | 0.0585 | 0.05878 | l/μmolN・day | Fujii et al. (2005), Sugimoto et al. (2010) |
| PL Potential maximum growth rate at 0℃ | $V_{0,PL}$ | 0.1 | 3.2 | 1.5 | 1.2 | 0.8 | 3.2 | /day | Shigemitsu et al. (2012) |
| PL Potential maximum affinity for $NO_3$ | $A_{0,NO3,PL}$ | 1 | 437 | 171 | 10 | 252 | 512 | l/molN・s | Shigemitsu et al. (2012) |
| PL Half satuation constant for $NO_3$ | $K_{NO3,PL}$ | 0.5 | 3 | 2.9194 | 3 | 3 | 3 | μmolN/l | Eslinger et al. (2000), Jiang et al. (2003) |
| PL Half satuation constant for $NH_4$ | $K_{NH4,PL}$ | 0.5 | 0.5 | 1.3129 | 0.3 | 0.3 | 2.3 | μmolN/l | Eslinger et al. (2000), Fujii et al. (2005) |
| PL Half satuation constant for $Si(OH)_4$ | $K_{SiL,PL}$ | 3 | 6 | 4.2857 | 6 | 6 | 6 | μmol/l | Yoshie et al. (2007) |
| PL Half satuation constant for FeD | $K_{Fed,PL}$ | 0.05 | 0.05 | 0.0887 | 0.09 | 0.1 | 0.2 | nmol/l | Coale et al. (2003) |
| PL Temperature coefficient for photosynthetic rate | $k_{PL}$ | 0.0392 | 0.0693 | 0.0392 | 0.0693 | 0.0693 | 0.0693 | /degC | Eslinger et al. (2000), Fujii et al. (2005) |
| PL Mortality rate at 0℃ | $M_{PL0}$ | 0.029 | 0.036941 | 0.034956 | 0.029 | 0.029 | 0.05878 | l/μmolN・day | Fujii et al. (2005), Yamanaka et al. (2004) |
| ZS Maximum rate of grazing PS at 0℃ | $G_{RmaxS}$ | 0.3 | 0.7933 | 0.3 | 0.31 | 0.4 | 4 | /day | Yoshie et al. (2007), Yoshikawa et al. (2005) |
| ZS Threshold value for grazing PS | $PS_{ZS*}$ | 0.04 | 0.364 | 0.364 | 0.043 | 0.043 | 0.364 | μmolN/l | Eslinger et al. (2000), Sugimoto et al. (2010) |
| ZL Maximum rate of grazing PS at 0℃ | $G_{RmaxL,PS}$ | 0.05 | 0.05 | 0.05 | 0.1 | 0.1 | 0.541 | /day | Eslinger et al. (2000), Fujii et al. (2005) |
| ZL Maximum rate of grazing PL at 0℃ | $G_{RmaxL,PL}$ | 0.135 | 0.251 | 0.135 | 0.49 | 0.4 | 0.541 | /day | Fujii et al. (2005) |
| ZL Threshold value for grazing PS | $PS_{ZL*}$ | 0.01433 | 0.043 | 0.043 | 0.04 | 0.04 | 0.043 | μmolN/l | Eslinger et al. (2000), Fujii et al. (2005) |
| ZL Threshold value for grazing PL | $PL_{ZL*}$ | 0.01433 | 0.043 | 0.018426 | 0.04 | 0.04 | 0.043 | μmolN/l | Eslinger et al. (2000), Fujii et al. (2005) |
| ZP Maximum rate of grazing PL at 0℃ | $G_{RmaxP,PL}$ | 0.1 | 0.4 | 0.1429 | 0.2 | 0.2 | 0.4 | /day | Eslinger et al. (2000) |
| ZP Threshold value for grazing PL | $PL_{ZP*}$ | 0.01433 | 0.043 | 0.018426 | 0.04 | 0.04 | 0.043 | μmolN/l | Eslinger et al. (2000), Fujii et al. (2005) |



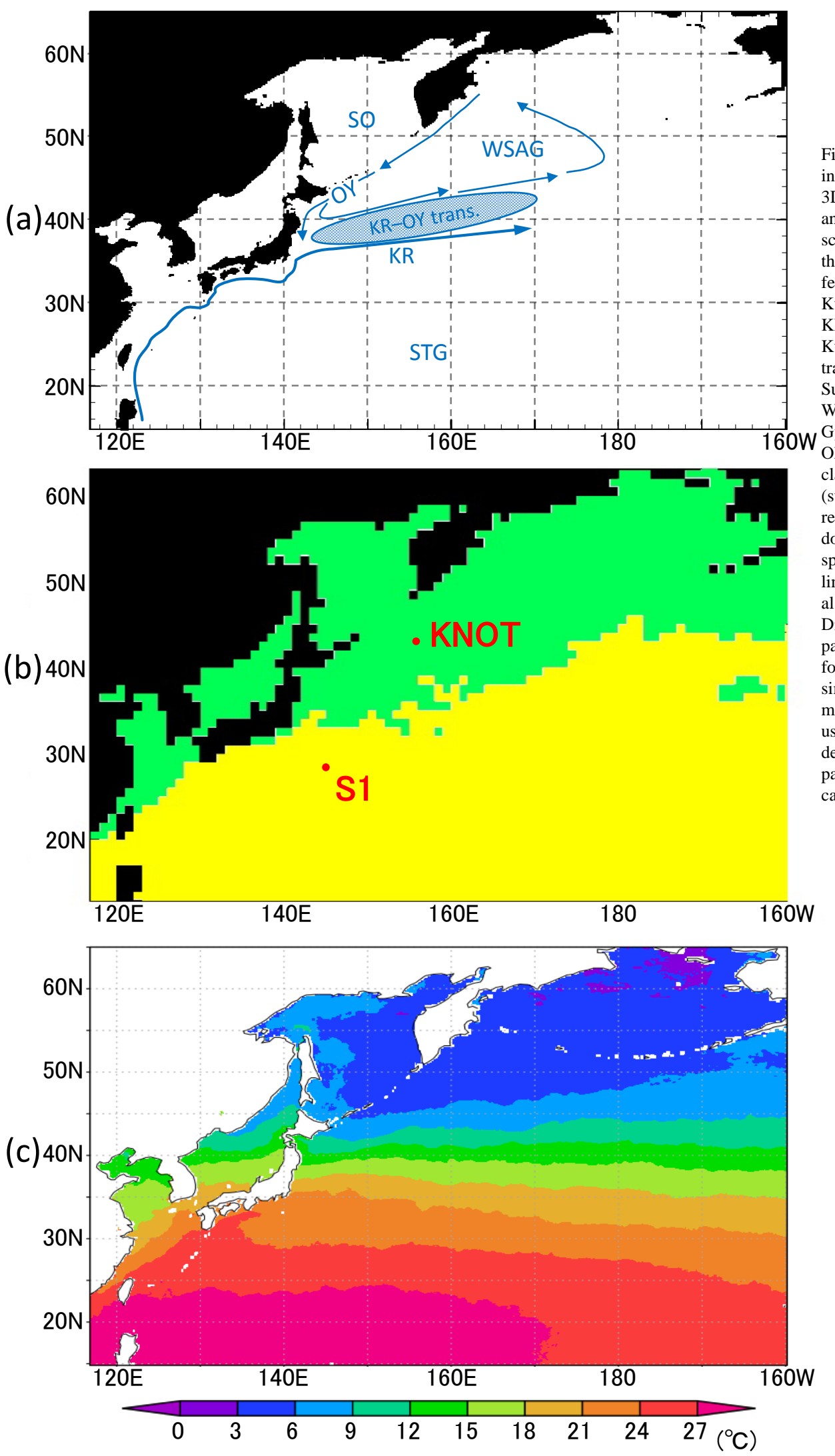

Figure 1. (a) Model domain in the WNP region of the 3D NSI-MEM. Blue arrows and symbols depict a schematic representation of the main circulation features in the WNP (KR: Kuroshio, OY: Oyashio, KR-OY trans.: the Kuroshio–Oyashio transition region, STG: Subtropical Gyre region, WSAG: Western Subarctic Gyre and SO: the sea of Okhotsk). (b) Two classified provinces (subarctic and subtropical regions) based on the dominant phytoplankton species and nutrient limitations by Hashioka et al. (in preparation). Different ecosystem parameters (Table 2) are set for each province in the simulation. (c) Annual mean SST of satellite data used for simulation of SST-dependent physiological parameters (SST-dependent case).

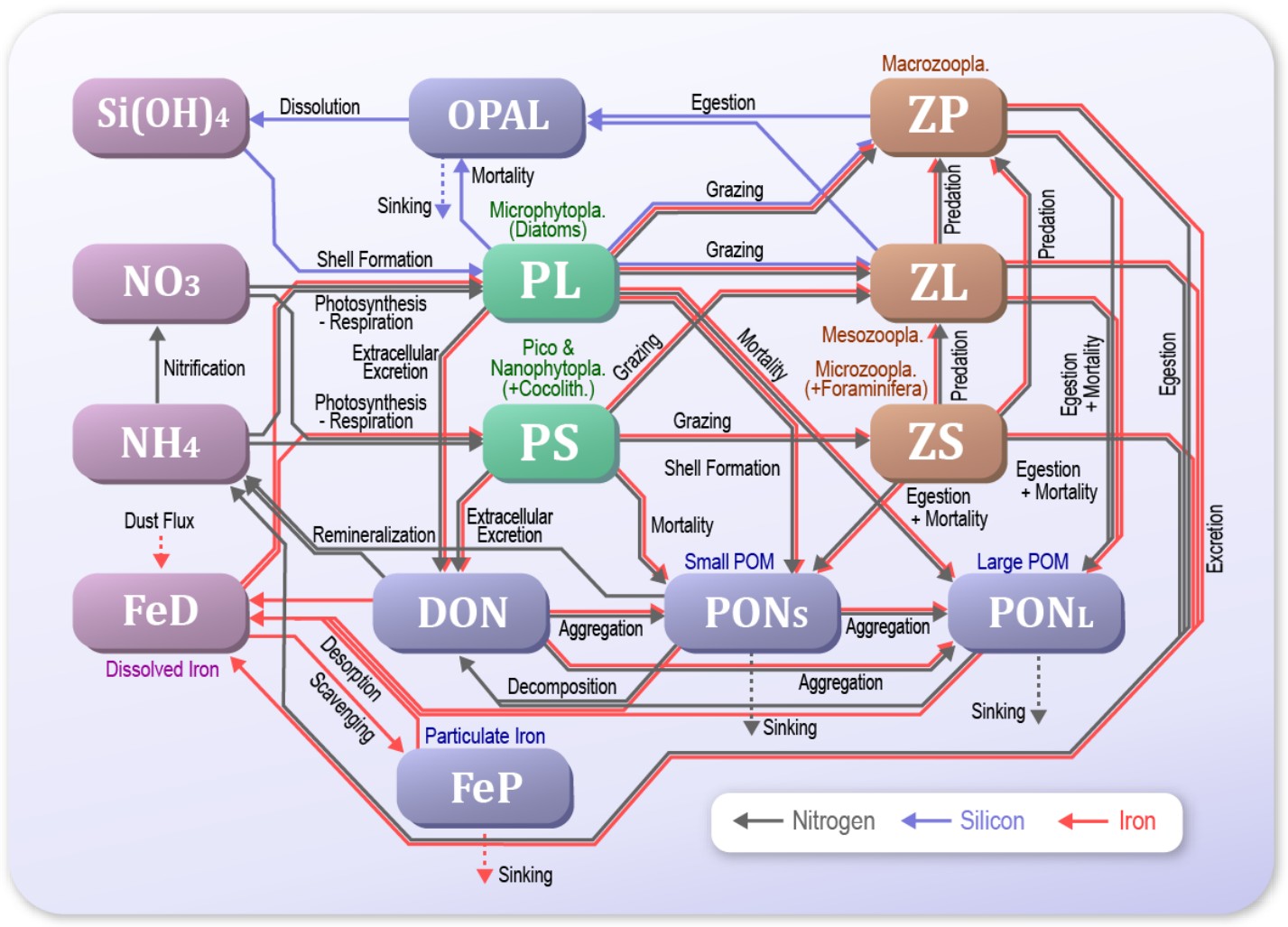

Figure 2. Schematic view of the NSI-MEM interactions among the fourteen components. Green colour boxes and brown boxes indicate phytoplankton and zooplankton, respectively. Blue boxes are particulate/dissolved matters. Violet boxes show nutrients and essential micronutrient.

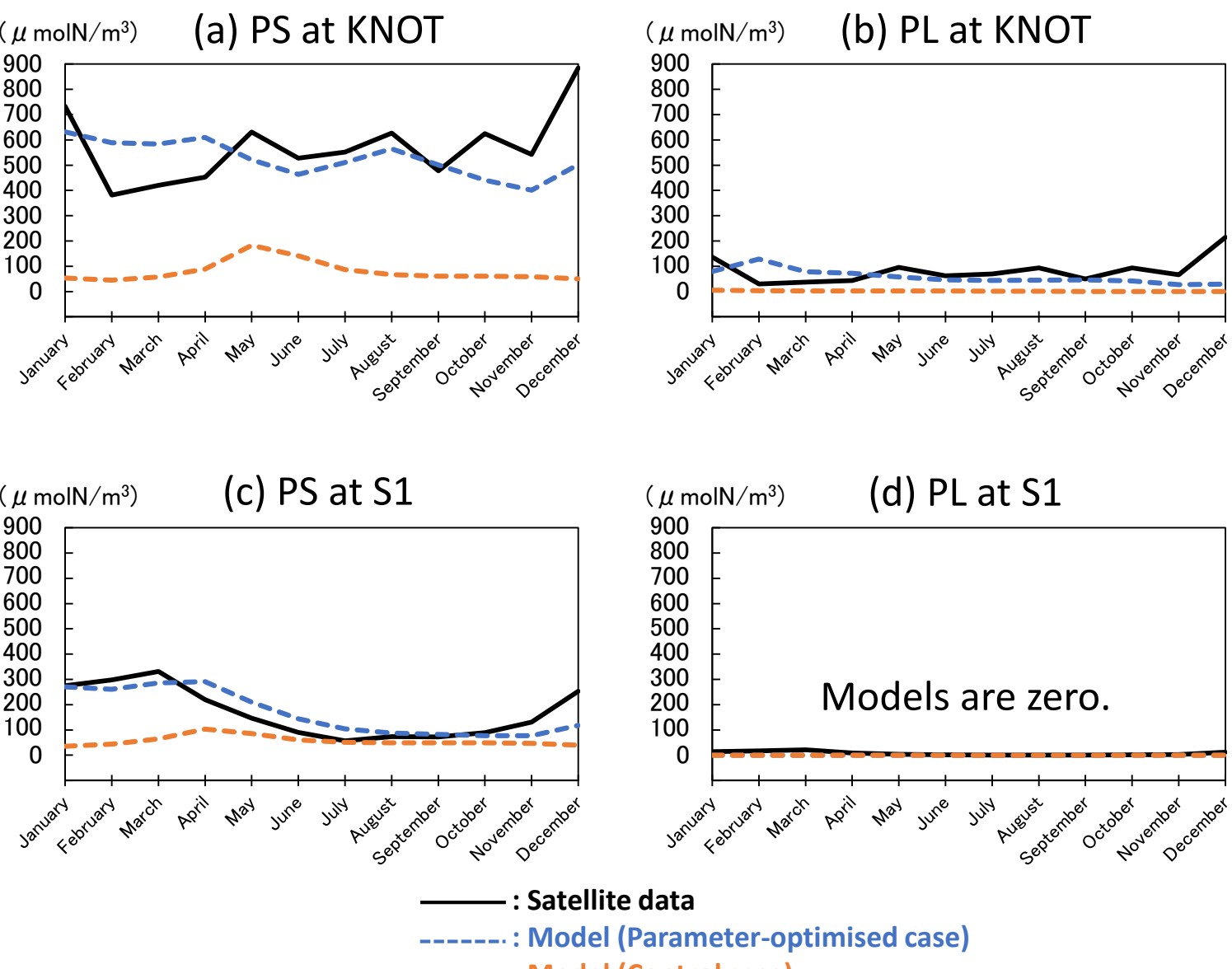

Figure 3. Seasonal variations of surface phytoplankton (PS: small phytoplankton and PL: large phytoplankton) biomass in the 1D NSI-MEM and satellite data at St. KNOT and St. S1 shown as typical observational points of the subarctic and the subtropical regions, respectively. (a) PS at St. KNOT, (b) PL at St. KNOT, (c) PS at St. S1, and (d) PL at St. S1 where the concentrations of the two model cases are almost zero, and that of the satellite is also remarkably small. The unit conversion between the simulation data (molN/m$^3$) and the satellite data (gchl-a/m$^3$) is referred to as the nitrogen-chlorophyll ratio of PL= 1: 1.59 and PS= 1: 0.636 (Shigemitsu et al., 2012). The same conversion of nitrogen-chlorophyll is used to Fig. 4, Fig. 6, Fig. 8 and Fig. 10.

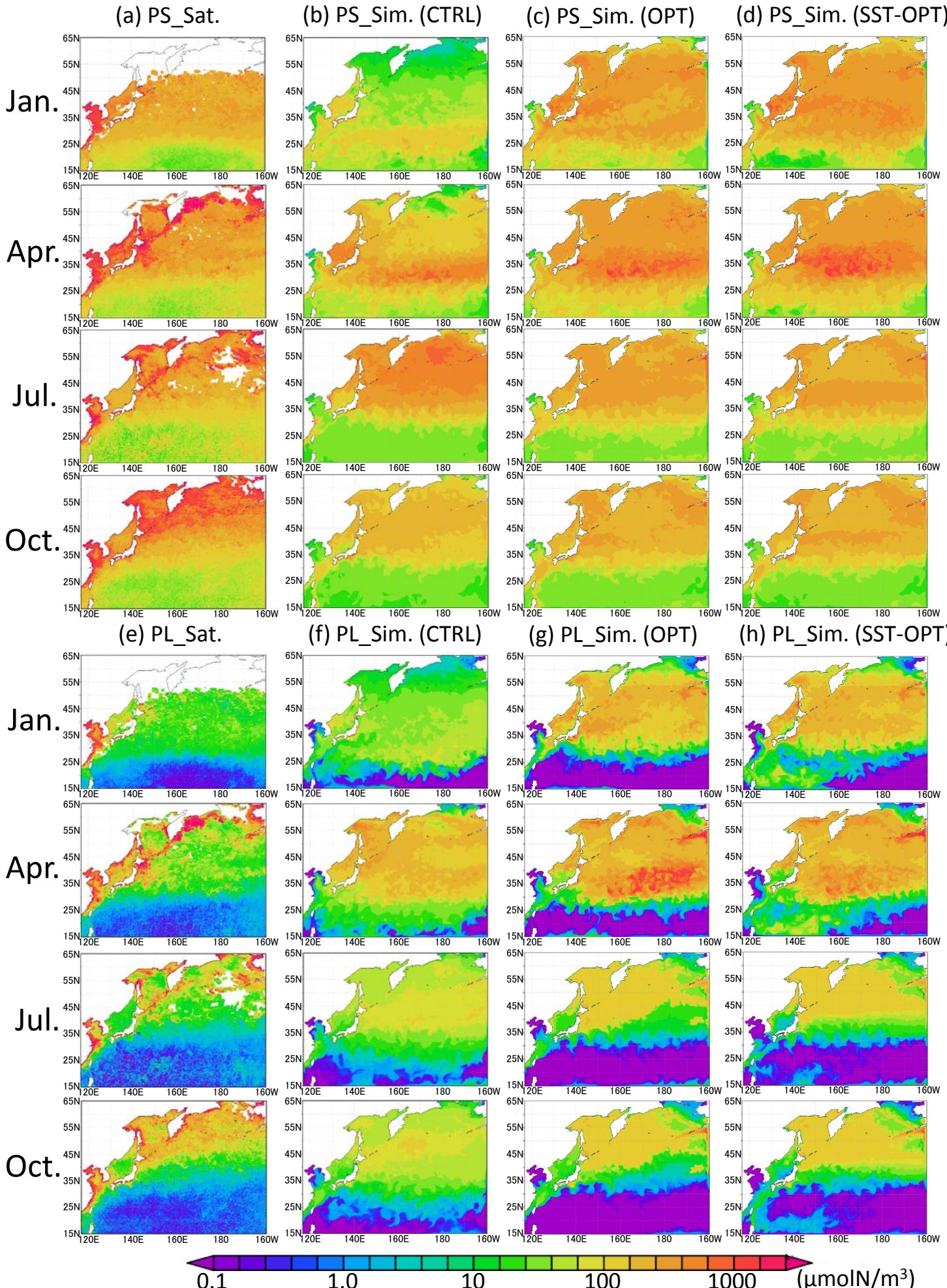

Figure 4. Horizontal distribution of phytoplankton at the surface in 1998. (a) PS (small phytoplankton) from satellites observations, (b) PS in Control case, (c) PS in the Parameter-optimised case, and (d) in the SST-dependent case. (e), (f), (g), and (h) are the same except for PL (large phytoplankton). Areas without satellite data are left blank.

## Control case

### (a) Lag correlation

### (b) Significance level

## Parameter-optimised case

### (c) Lag correlation

### (d) Significance level

## SST-dependent case

### (e) Lag correlation

### (f) Significance level

Figure 5. Horizontal distribution of lagged (within ±2 months) correlation coefficients for the monthly time series of phytoplankton (PL+PS) concentration between the simulation and the satellite data in each grid at the surface in 1998, and the significance levels. (a, b) Control case, (c, d) Parameter-optimised case, and (e, f) SST-dependent case. Areas less than the number of seven monthly mean satellite data and in the coastal regions where the bottoms are less than 200 m are left blank.

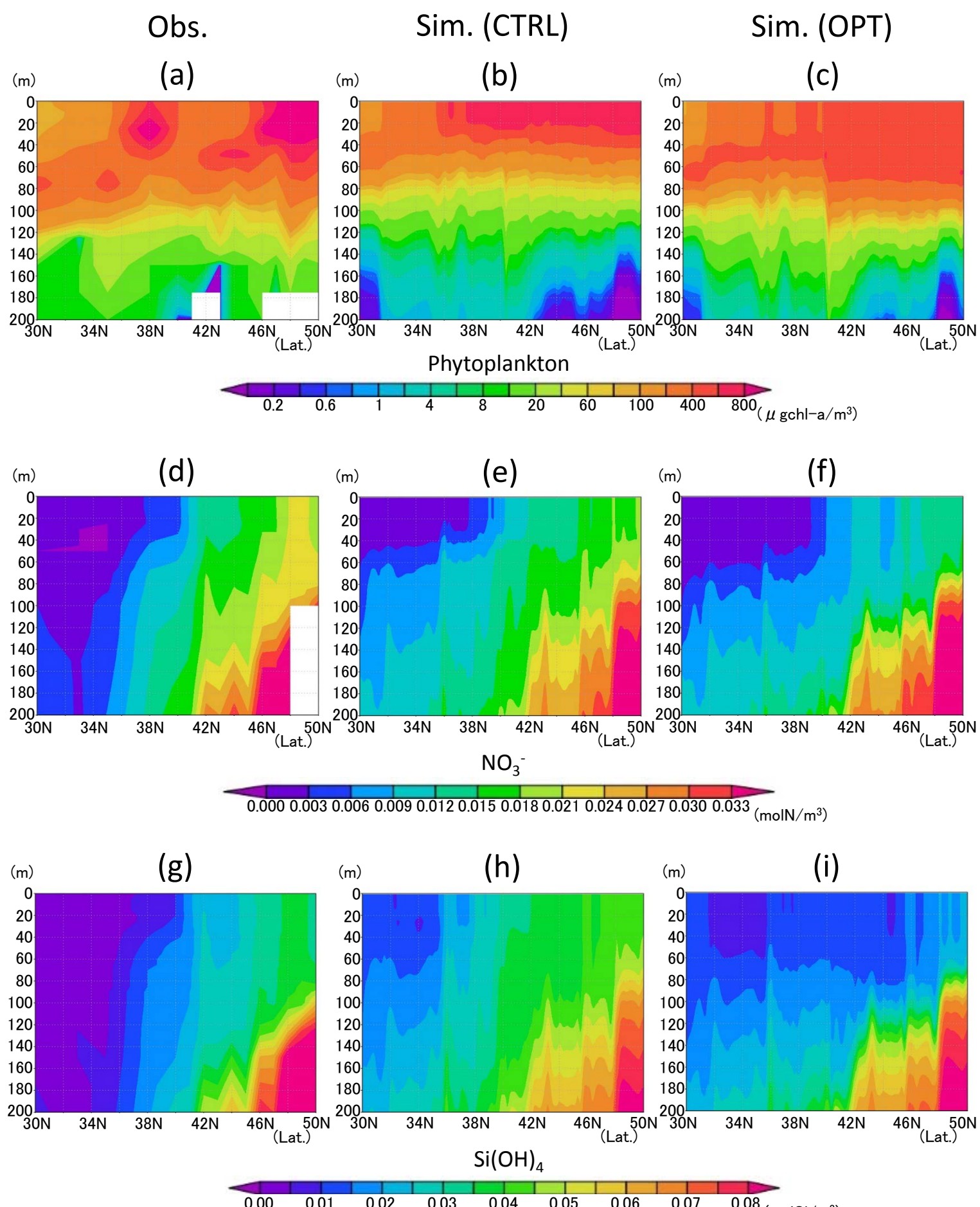

Figure 6. Vertical distribution of phytoplankton (a, b, c), nitrate (d, e, f) and silicate (g, h, i) along the $165^\circ$ E section in June 1998. (a, d, g) Data in situ observed during 16th June to 21st June in 1998 downloaded from World Ocean Database 2013. (b, e, h) Simulation result of Control case in June 1998 mean. (c, f, i) Simulation result of Parameter-optimised case in June 1998 mean. Areas of missing values are left blank.

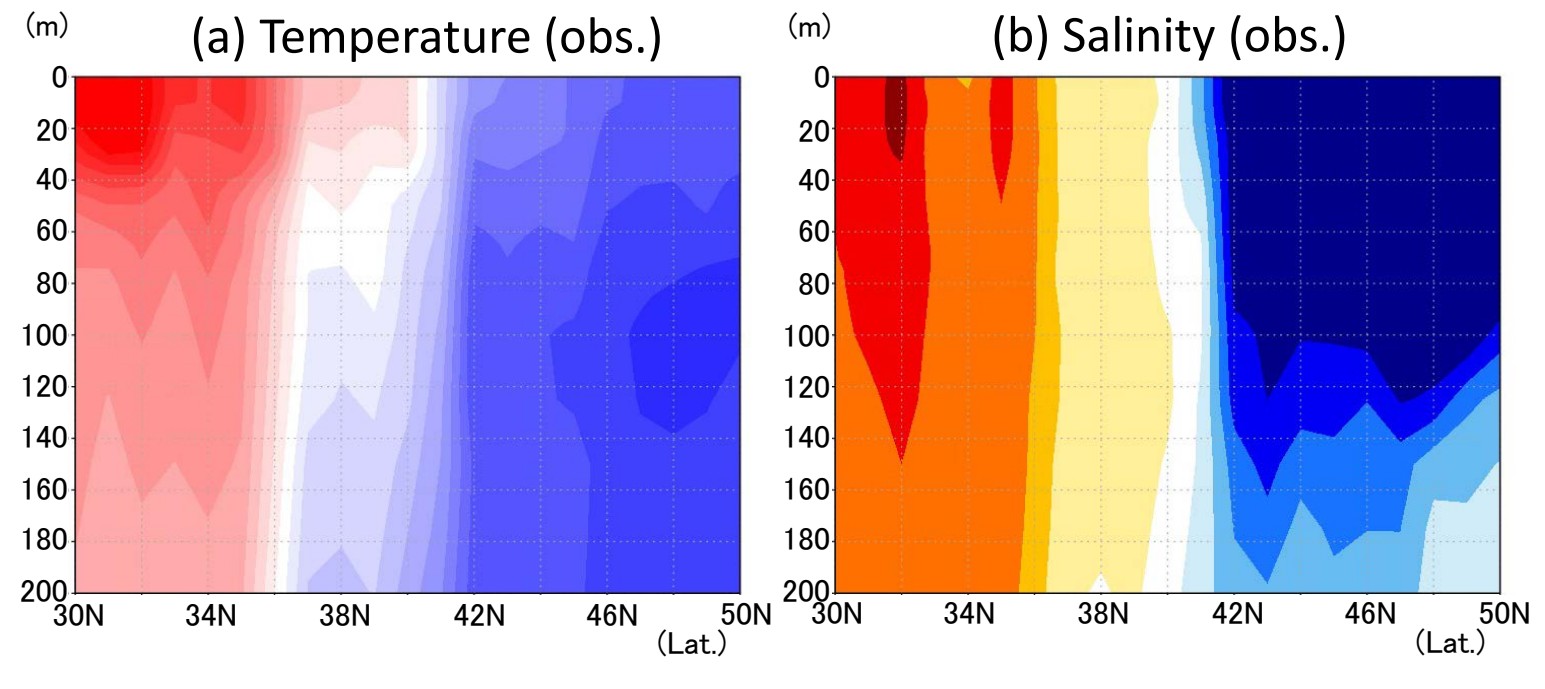

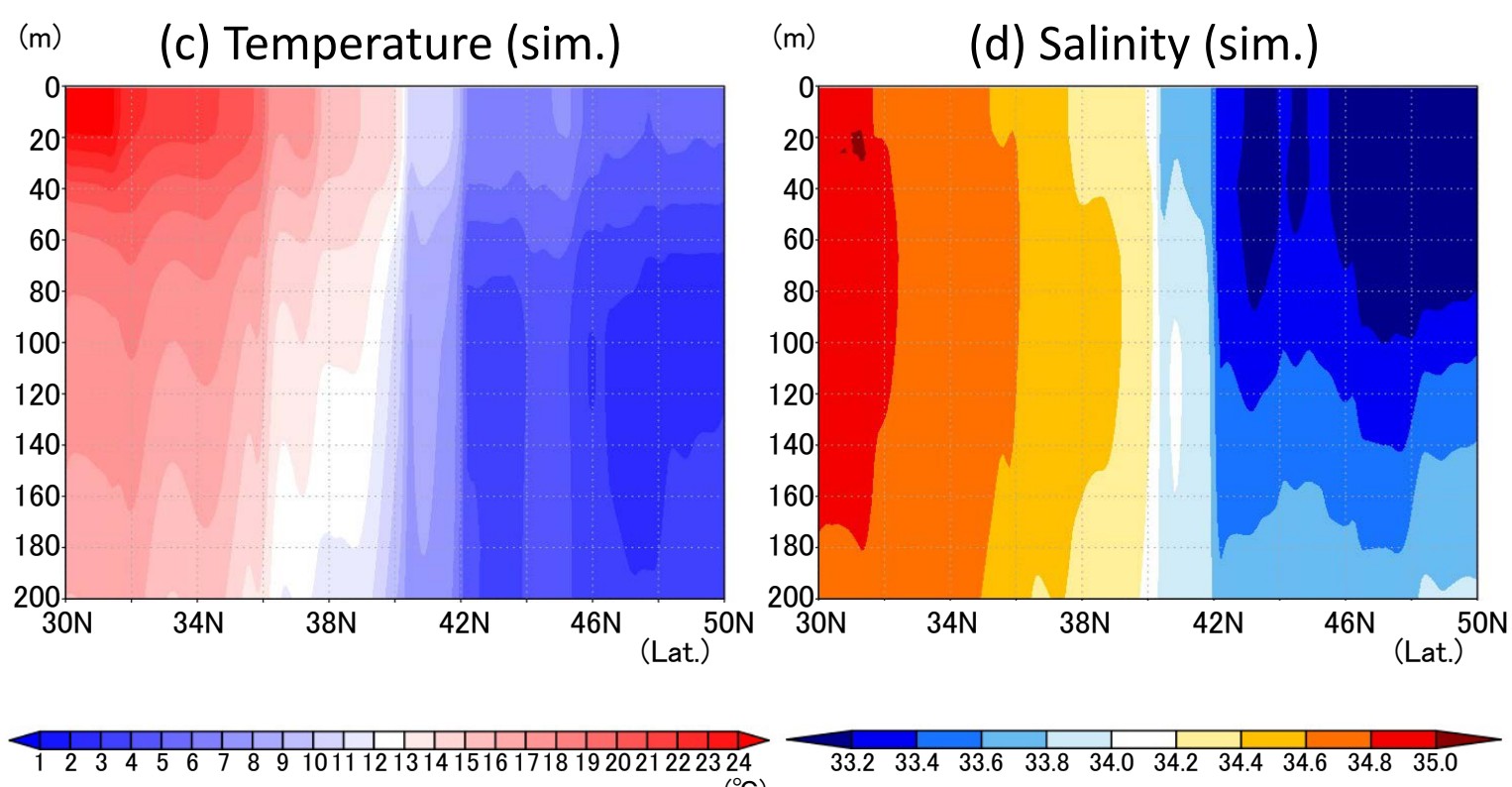

Figure 7. Vertical distribution of temperature (a, c) and salinity (b, d) along the 165° E section in June 1998. (a, b) Data in situ observed during 16th June to 21st June in 1998 downloaded from World Ocean Database 2013. (c, d) Physical field in June 1998 mean used in the 3D NSI-MEM.

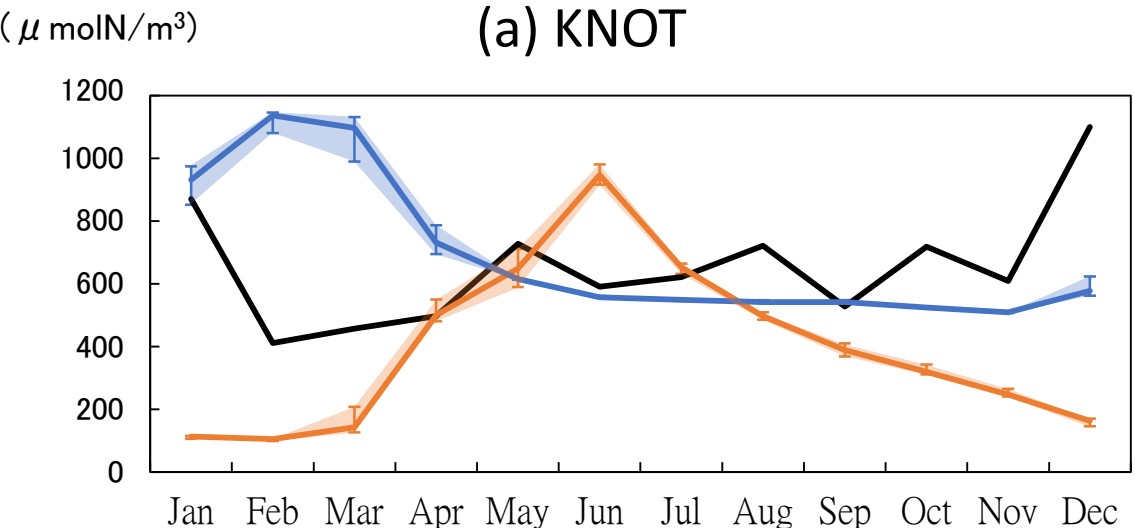

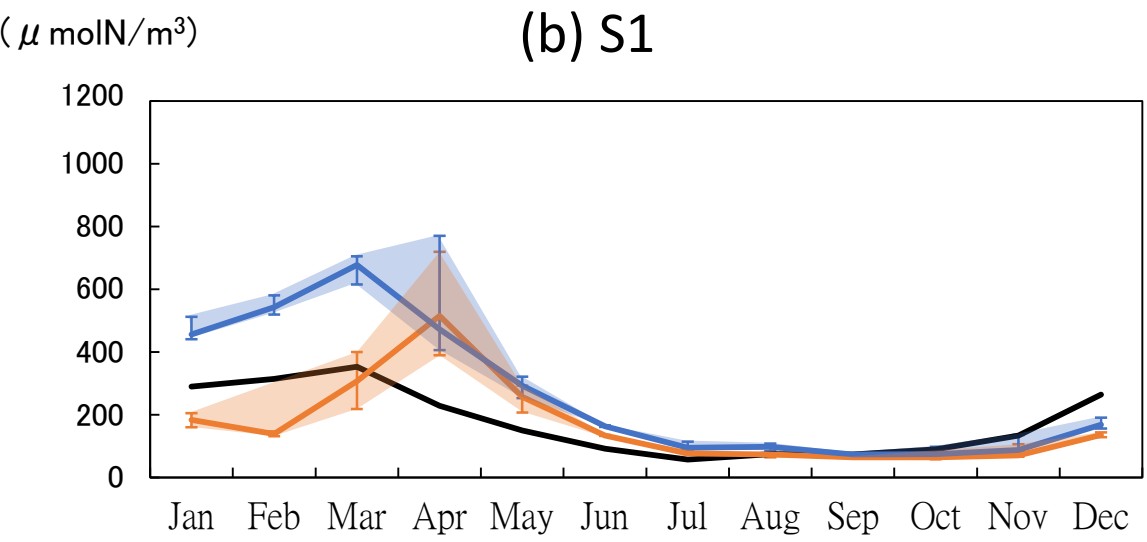

_______ : Satellite data

_______ : Model (Parameter-optimised case)

_______ : Model (Control case)

Figure 8. Time series of phytoplankton (PL+PS) concentration in the 3D NSI-MEM and satellite data at (a) St. KNOT and (b) St. S1. Error bars and shade of the simulations show the maximum and minimum values in $\pm 0.3°$ around the grids of St. KNOT and St. S1.

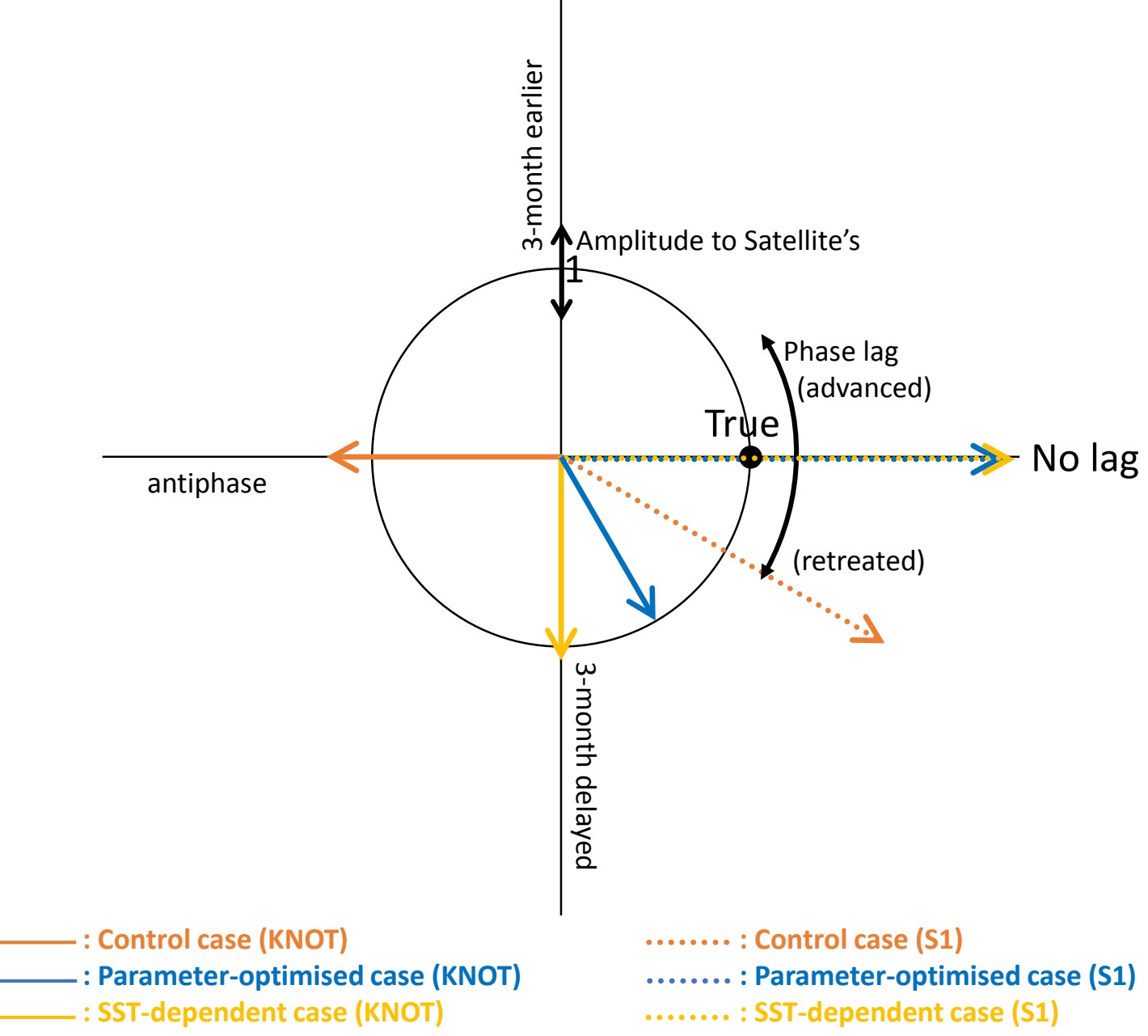

Figure 9. Diagram showing the amplitude and the phase of seasonal variations in the three model cases compared with those in the satellite data. Based on the seasonal variation in the satellite data, the radius indicates the relative amplitude (model/satellite) of seasonal variation for each model case and the angle from the positive x-axis shows the time lag of the maximum concentration for each model case (i.e., the point (1, 0) shown as 'True' is a match within 1 month and 30 degrees error range to the satellite data). The blue dotted line (Parameter-optimised case at St. S1) and yellow dotted line (SST-dependent case at St. S1) overlap on the no-lagged x-axis.

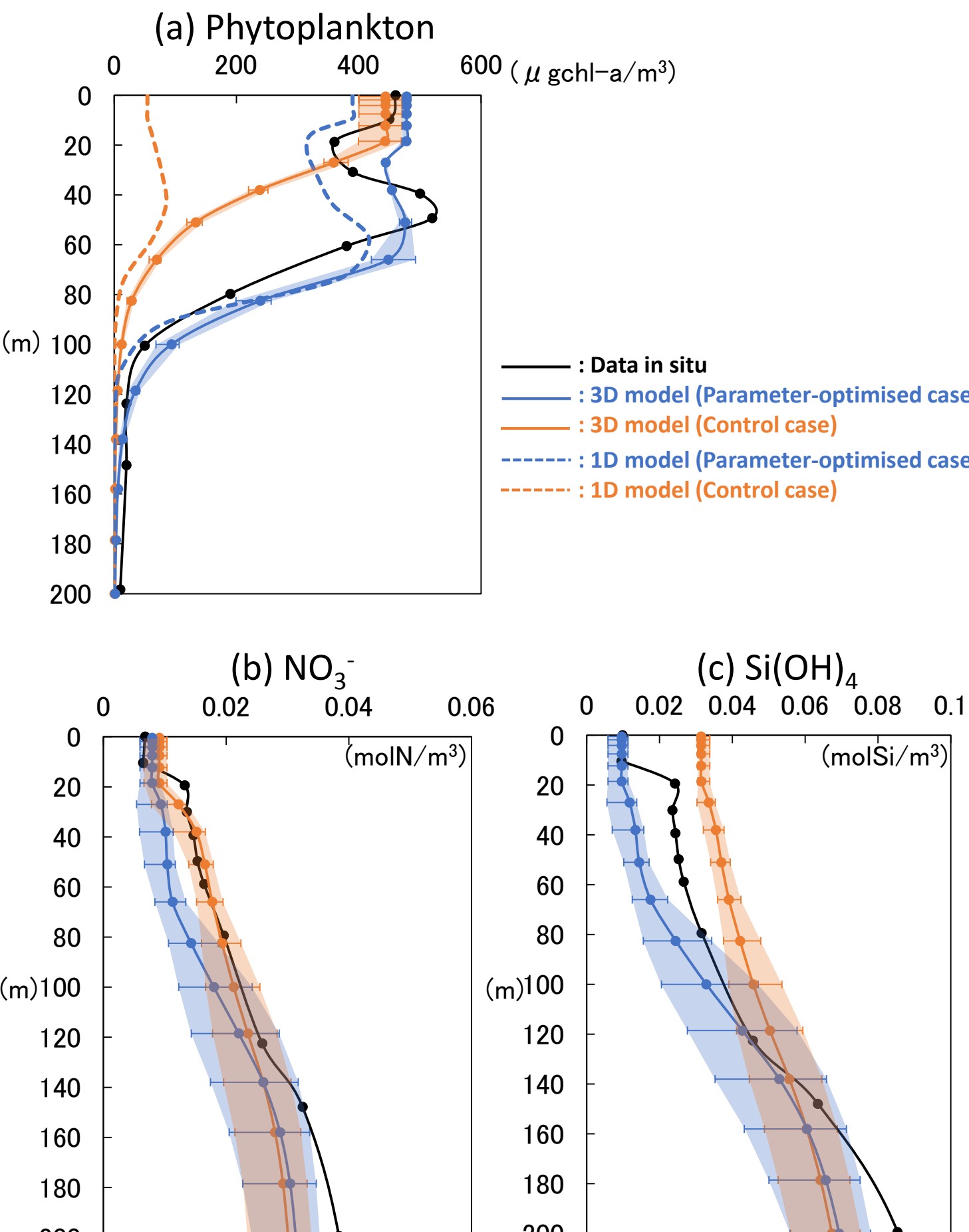

Figure 10. Vertical distributions of (a) phytoplankton (PL+PS) from the 3D model (solid line), 1D model (dashed line) and in situ data, (b) nitrate and (c) silicate concentrations from the 3D model (solid line) and in situ data at St. KNOT on 20th July, 1998. Error bars and shade of the 3D simulations show the same mean as those of Fig. 8.

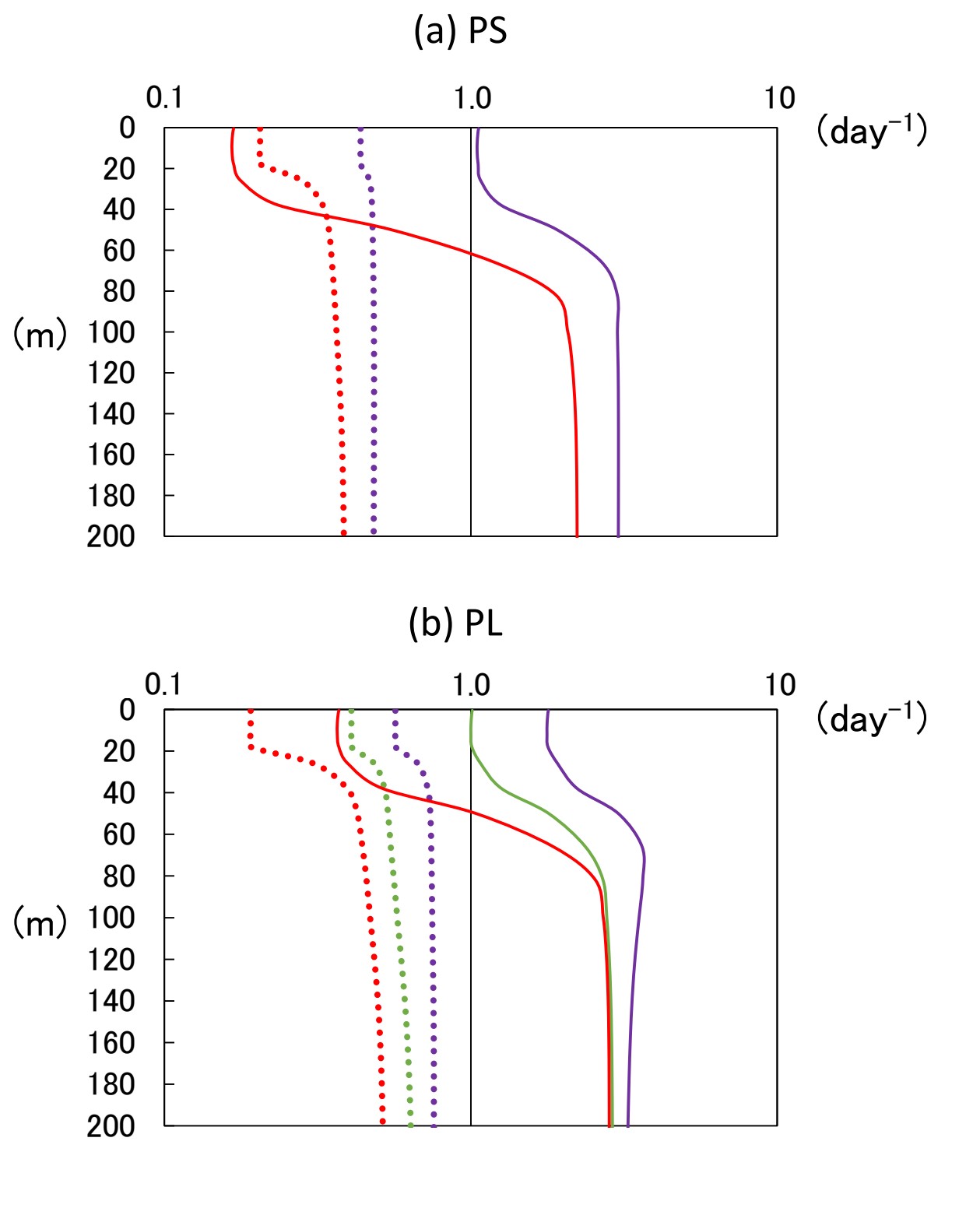

Figure 11. Vertical distributions of limited growth rates by nitrogen, silicate and dissolved iron simulated from the 3D model of (a) PS and (b) PL at St. KNOT on 20[th] July 1998. The smallest rate of dissolved iron most heavily limits the rate of phytoplankton's photosynthesis. These limited growth rates ($molN/m^3/day$) were divided by the PS or PL biomass ($molN/m^3$) to standardize.

## (a) Control case

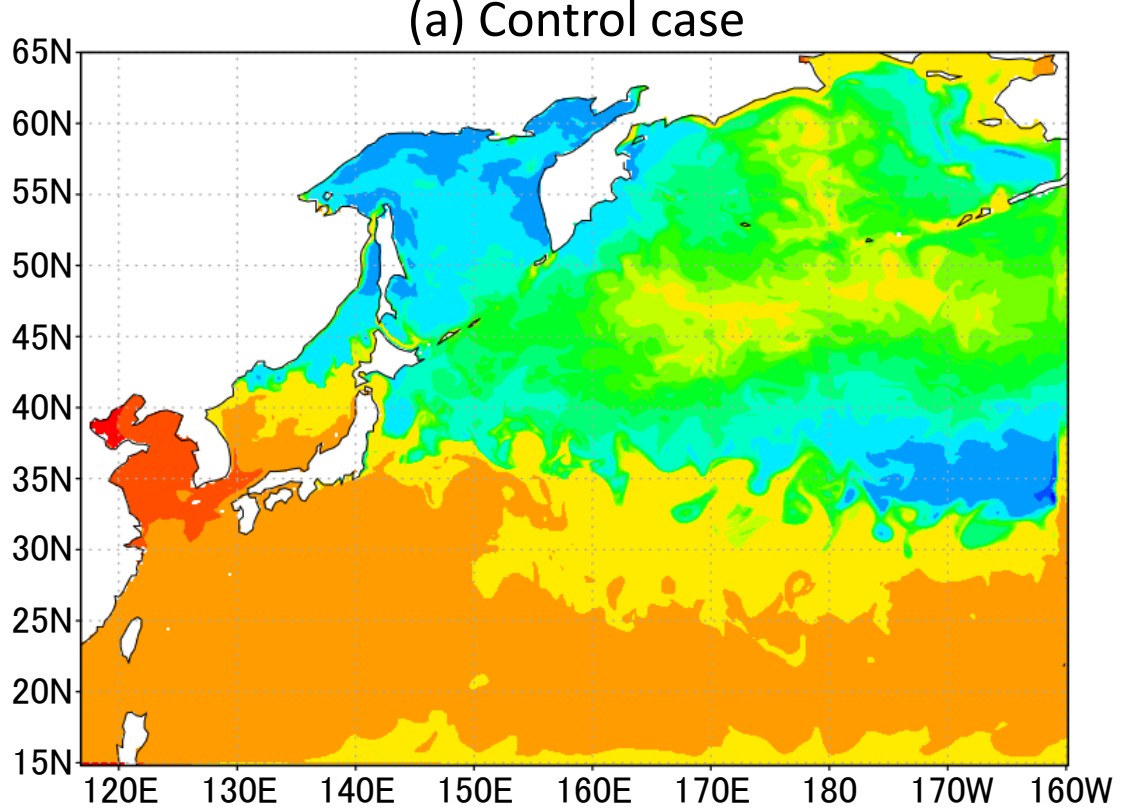

## (b) Parameter-optimised case

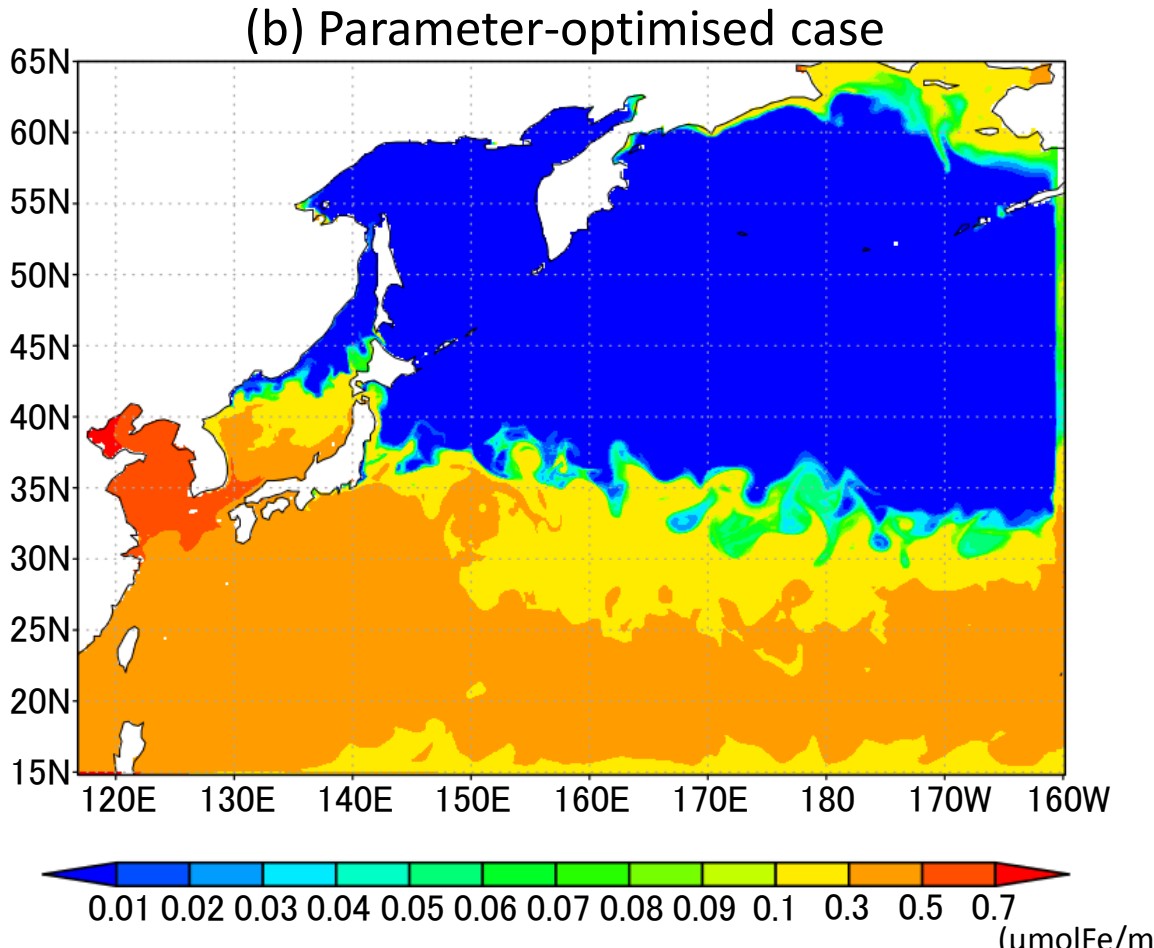

0.01 0.02 0.03 0.04 0.05 0.06 0.07 0.08 0.09 0.1 0.3 0.5 0.7

(μmolFe/m³)

Figure 12. Horizontal distribution of dissolved iron in the surface seawater layer for July 1998; (a) Control case and (b) Parameter-optimised case.