# Peer review of "Biological data assimilation for parameter estimation of a phytoplankton functional type model for the western North Pacific"

_Ocean Science, 2017_

## Referee Comment (RC1) · Anonymous Referee #1 · 21 Jun 2017

Review comments on "Biological data assimilation for parameter estimation of a phytoplankton functional type model for the western North Pacific" by Hoshiba et al.

1. Summary

The authors conducted numerical simulations of a lower trophic level (LTL) ocean ecosystem model for the northwestern Pacific Ocean. To simulate a realistic temporal evolution of phytoplankton biomass, the authors employed a 1-D ecosystem model and optimized 23 ecosystem model parameters by a micro-genetic algorithm. The optimization was performed in two specific locations in the northwestern Pacific Ocean; one is the subtropical region (Station S1), the other is the subpolar region (Station

[Figure]

KNOT). The optimized model parameters were applied to a LTL 3-D ecosystem model simulations in the northwestern Pacific Ocean, for which a physical field obtained from an eddy-resolving ocean model were used by an off-line technique. The authors performed 3 different simulations by the 3-D ecosystem model: a simulation with an unoptimized ecosystem model parameters, a simulation with the optimized parameters for the subtropical and subpolar regions, and a simulation with the optimized parameters varying with the sea surface temperature from the subtropical to the subpolar region. Based on these experiments, the authors reported that they successfully improved the seasonal variation and vertical distribution of phytoplankton biomass.

2. General comments

It is interesting to see how the state-of-art LTL ecosystem model simulates the realistic temporal and spatial evolution of plankton biomass, since it constitutes an important part of the geochemical cycles simulated by a model. The LTL ecosystem model used for the current study is the one including iron cycle and its interaction with biomass distribution, which is considered to be necessary to improve the biomass distribution simulated by a model in high nutrient low chlorophyll (HNLC) regions such as the northwestern Pacific Ocean. I think the basic strategy of the current work is well-considered and reasonable for a step-by-step improvement of the LTL ocean ecosystem model - i.e., tune the model parameters by a 1-D model which is computationally inexpensive, and then apply the tuned parameters for a 3-D ecosystem model and examine how the spatial distribution of plankton biomass is simulated. However, it looks to me that the tactics employed in the study is not necessarily suitable for the purpose of the study as described below. The authors did not obtain reasonable improvement of simulated biomass by the 1-D optimization experiments, probably due to an inappropriate choice of tuning parameters, and failed to simulate realistic biomass distribution in the study area. Since the basic strategy is reasonable, I would recommend the authors to rework this issue addressing the following points.

3. Major points

[Figure]

- I would say that the result obtained from the 1-D parameter optimization is quite miserable and frustrating. If I understand the setup of the experiment correctly, the cost function is composed of only 24 values (12 months × 2 types of phytoplankton biomass), while the number of optimization parameters is 23. This is mathematically equivalent to optimizing 24 modeled values by 23 model parameters. For such an experiment design, one can expect nearly perfect fit of the modeled values to the observation, if the model appropriately describes the process concerned and the choice of the tuning parameter is appropriate. Nevertheless, the authors failed to fit the modeled phytoplankton biomass to the observed ones, nor even reproduce annual mean phytoplankton biomass: the simulated PS biomass at the subtropical station S1 is nearly 10 times larger than the observed one; the simulated PL biomass at the subpolar station KNOT is more than 5 times larger than the observed one during the winter period (Fig. 3), both of which are essential to describe the dominant species of phytoplankton in each area. To be honest, I do not find any benefit to apply the parameter sets, which provides such a large discrepancy from observation, for 3-D model simulations. I would recommend the authors to redo the optimization experiments by taking the points described below into account.

- How did the authors select the parameters used for the optimization? It looks to me that the authors selected a number of physiological parameters for the modeled phytoplankton, while did not select any parameters describing physical processes of the system (e.g., sinking rate, etc). As shown in Fig. 3b, the model exhibits too much PS biomass throughout the year, indicating nutrients in the euphotic zone are repeatedly recycled without being extracted from the system, probably due to insufficient parameterization for sinking/scavenging processes. This situation can be also seen in Fig. 7 - the discrepancy between the modeled and observed NO3 becomes larger after optimization (the unoptimized parameter set gave more reasonable result). Therefore, my recommendation is to reconsider the selection of tuning parameters, so as to tune the nutrient input to/output from the euphotic zone, and examine the improvement of nutrient distribution before examining modeled phytoplankton biomass. Otherwise the
derived optimal physiological parameters are not describing the realistic function of phytoplankton physiology.

- The current work did not take the advantage of the physical field obtained from the high resolution ocean model with data assimilation. During the last two decades, 3D ocean ecosystem models have been suffered from insufficient descriptions of physical fields obtained from low-resolution ocean models (e.g., reproducibility of mixed layer depth, its spatial distribution and seasonal variation, location of subtropical-subpolar boundary, coastal upwelling, etc.). Although the current study utilizes a physical field which is supposed to be free from such problems, I cannot find any distinguishable improvements compared to the studies using the low-resolution physical field. This is probably due to an insufficient implementation of nutrient cycles in the model, particularly the process describing input/output of nutrients into/from the system. My recommendation is, first of all, optimize the modeled nutrient cycles before tuning, examining or discussing the physiological parameters of the LTL ecosystem model. As far as I know, the coauthors listed here have done a number of excellent works and should have sufficient knowledge and experience for this.

- Description for the experiment design is insufficient. I think more description is necessary for a further review (as described in specific points).

4. Specific Points

- Line 24-64: Although the authors provided a concise review in introduction, the papers cited here seems to be largely biased toward the papers written by the co-authors of this work. I think it would be one more advantage of this work, if the authors mention the work by other groups.

- Line 33: vales –> values?

- Line 45-49: The construction of the sentence seems strange.

- Line 71-72: The authors described that the physical field used for the 3D experiment

is obtained from a 3D-Var data assimilation, in which temperature (T), salinity (S) and sea surface hight (SSH) are assimilated. This means increments of T, S and SSH are added to the analysis field in each analysis time step. Is the physical field satisfy the mass conservation after the SSH assimilation? If not, how much amount of artificial sink/source of passive tracers should we expect? Is it not essential for the LTL ecosystem model simulation, particularly close to the sea surface?

- Line 76-77: What does "similar" mean? The authors should describe the difference from the cited work.

- Line 76-77: How did the authors provide dust flux for dissolved iron? The earth system model (Watanabe et al., 2011) contains iron in the dust? If not, how did the authors define the amount of iron concentration in the dust flux? Description is needed.

- Line 78: How did the authors define nutrient supply from the river? Does the CORE-2 provide nutrient concentration in river run-off? If not, how the author defined the value?

- Line 78: How did the authors define the nutrient supply from sediment over the shelf area?

- Line 79-80: This sentence needs citation.

- Line 68-83: Where does the iron in the system come from? Many studies addressed the importance of iron supply from the Sea of Okhotsk to the northwestern Pacific Ocean. How did the authors describe the iron supply from the Sea of Okhotsk?

- Line 68-78: How did the authors define the initial condition of nutrient distribution (nitrate, silicate and iron)? Description is needed.

- Line 79: Describe the range of restoring boundary layer and the restoring time scale.

- Line 80-82: How about the seasonal cycle of mixed layer depth and its spatial distribution? Is it well reproduced? I'm asking this question because I believe the mixed layer depth is the most important factor to regulate the nutrient supply into the euphotic

zone.

- Line 82-83: Are the nutrients in an equilibrium state after 1985-1998 integration? Is the nutrient distribution of the equilibrium state consistent with observations (e.g., Wold Ocean Atlas)?

- Line 88-89: The construction of the sentence seems strange.

- Line 89-90: What does the "similar" mean? The authors should describe which parameter(s) had been changed from Shigemitsu et al. (2012).

- Line 90: I suggest to use 'control-case' instead of 'default case'.

- Line 91-97: This experiment design is interesting, while I would suggest the authors to explain the basic philosophy behind this. It looks to me that introducing temperature dependency on many physiological parameters of the LTL ecosystem model is equivalent to rewrite the governing equation drastically, since some of the phytoplankton model parameterization already involve temperature dependency.

- Line 100-104: Description for the temporal and spatial resolution of the data is needed.

- Line 105-107: Why the authors used AVHRR data for SST-dependent case, instead of using SST obtained from the physical model? I think this experiment design may introduce a discrepancy: the modeled phytoplankton is controlled by two different temperatures (one is from the physical model and the other is from AVHRR). If I'm wrong, please explain which temperature is used to calculate the modeled phytoplankton biomass.

Line 114: A description for the selected parameters is necessary. The names of the selected parameters seem to be the same with the definitions in Shigemitsu et al. (2012), while it is not clear to the most of (potential) readers what do they mean. I suggest to implement a short description for each parameter in Table 2.

- Line 115: Again, what does the "similar" means? The difference should be described.

- Line 119-120: The construction of the sentence is strange, and I cannot really understand the meaning of the sentence. Does the sentence mean "the parameter set which provides the lowest cost is reserved"?

- Line 125: Why should the number of the population used in the genetic algorithm optimization be the same with the number of tuning parameters?

- Line 131: Why do the authors used the same weights for PS and PL? Is it based on uncertainties of satellite-derived biomass?

- Line 139: 'too small' –> 'smaller than the prescribed threshold'.

- Line 118-116: If I understand correctly, the parameter optimization by the 1-D model used a 1-year time window. Why the authors do not use a longer time window for the optimization? If the authors define the cost function by a multi-year window, the cost is more reliable. The computational cost is not essential in this case.

- Line 150-153: It looks to me that the following two sentences contradict each other; "the PS biomass was larger than the PL biomass at both St. KNOT and St. S1," and "Moreover, diatom, represented as PL, are a major group in the subarctic region.". Isn't it?

- Line 154-155: How do the authors evaluate the uncertainty of the biomass derived from satellite data?

- Line 149-159: Why does the optimized case exhibit such a large discrepancy from the satellite data? As I mentioned in the major points, I guess the selected parameters are not relevant to improve the discrepancy.

- Line 149-159: How much (percentage) is the reduction of the cost compared to the 'default case'? I cannot believe that the 'parameter-optimised case' for S1 gives smaller cost than the 'default case', since the PS biomass exhibits such large discrepancy from

the satellite data. Please show the total cost for 'default case' and 'parameter-optimised case', and cost for each month (e.g., a figure with the same abscissa as Fig. 3, while the ordinate is defined by cost for each month).

- Line 170-172: I'm a bit skeptical to the specification provided here. The authors employed a physical field obtained from an eddy-resolving model, yet they argued that the lack of the small-scale mixing is still responsible for the low-biased biomass close to the cost (or over the shelf). If this is true, what is the advantage to use the eddy-resolving physical field in this study? Since Fig. 4 and 9 successfully reproduced an eddying physical field, I guess a lack of nutrient supply from the seabed is a likely reason for the low-biomass close to the cost. How did the authors implement nutrient flux from seabed? Is it suitable to reproduce nutrient cycle over the shelf and/or close to the cost?

- Line 181-194: I think Fig. 6 (and associated analysis provided here) is an useful measure to characterize the performance of a LTL ecosystem model. But I would say, due to the large discrepancy between observed and simulated biomass (Fig. 3, 4 and 5), the analysis is not necessarily useful. I suggest to redo the analysis after a re-optimization of the model parameters as described in the major points.

- Line 196-203: How did the authors take into account the spatial and temporal representativeness of the modeled phytoplankton (and nutrient) for the comparisons? Since the horizontal distribution of the modeled properties has an eddy-scale fluctuation (e.g., Fig. 4), a direct comparison with in-situ data is meaningful, if and only if the physical model accurately reproduced the location and evolution of respective eddies. I'm not sure this is the case or not, since the physical model assimilated SSH (the reproducibility of realistic eddy fields depends on the spatial and temporal resolution of the assimilated SSH and the assimilation interval). If the location of respective eddies are not necessarily realistic, a mean value should be used for the comparisons (and standard deviation of the field should be used for the measure of uncertainty). Otherwise, we cannot argue which line in Fig 7 is closer to the in-situ data.

- Line 204-218: I'm also skeptical to the usefulness of the analysis and discussions provided here, since the background field for the modeled LTL ecosystem (i.e., nutrient fields) are not thoroughly examined nor confirmed to be realistic. The authors compared the vertical distribution of NO3 between model and observation only at one location. I think it is necessary to check the reality and weakness of the modeled nutrient fields before proceeding analyses for the modeled phytoplankton physiology. I suggest to compare the spatial and vertical profiles of nutrient fields (nitrate, silicate and iron) with available atlas and data sets.

- I found a number of strange construction of sentences. I think a consultation of the English sentences is necessary.

---

## Referee Comment (RC2) · Anonymous Referee #2 · 26 Jun 2017

Parameters estimation in biogeochemical models is a critical challenge and assimilation of data is an important tool to better constrain these parameters. This manuscript is thus dealing with a topic of relevance. It is clearly written and presented, figures are of good quality. I have however important criticisms concerning the methodology used in the manuscript. Mainly, I find that the paper does not provide enough convincing arguments that the way data assimilation is used here clearly improves model performances. The use of data assimilation requires a thorough quantitative assessment of model performances before assimilation and after assimilation and this assessment is insufficient here. Only the seasonal cycle of monthly averaged surface concentration of phytoplankton is presented and broadly compared with observation and this is

not sufficient. I would recommend that the authors strongly improve that point. Then, I would like that the authors comment of the propagation of errors that may happen when 1) interpolating a global chlorophyll product to their grid (is there not a regional product available for the region? ), 2) converting global chlorophyll to PFT, 3) converting chlorophyll to nitrogen values. The paper does not convincingly demonstrate that model performances are really improved thanks to the assimilation of such data (conversely looking at e.g. figure 3, we have the feeling that data assimilation degrades model performances see my details comments below). The readers would be better convinced by the gain obtained by the assimilation of such data if the authors presents a thorough error assessment computing error statistics like bias, RMS,. . . Therefore, I find that the paper cannot be accepted for publication in its present form and would require a substantial revision.

Detailed comments: Line 16-17: The authors say " The approach used a one-dimensional emulator that referenced satellite data". Please clarify what you mean by "emulator" and "reference to satellite data". I guess that you mean that the estimation of parameters is based on a 1D simulation using satellite data for constraining biogeochemical parameters.

Lines 17-18: The comparison with other models that do not assimilate data is only limited to the model used in this paper and cannot be considered as a general feature. Please reformulate.

Line 33-35: "Physiological parameters have often been tuned up empirically and arbitrarily, although ecosystem models have recently added more parameters to increase the number of prognostic and diagnostic variables". It is not clear here why there is an opposition of the two parts of the sentence (use of although). The fact that the number of models parameters increases makes it more and more difficult to tune them automatically. On the other, I agree that ecosystem models are more and more complex with the aim to increase their reliability but the increased number of uncontrolled parameters may make them less realistic.

Line 36 : Please clarify what you mean by Âń a reasonable estimate of the physiological parameters Âż Line 40: set and not set?

Line 42: I find that the use of "This algorithm"" may be confusing. I guess that here the authors refer to the model NSI-NEM and so I would use "model" instead of "algorithm".

Line 45: I guess that the sentence is incomplete. I would add something like 'based on " before following the . . .

Lines 44-49: This part would be better placed in the section on materials and methods

Line 50 : "and the Parametres-optimized approach" this is vague I would say and a micro genetic algorithm to optimize..

Lines 55-60: I would suggest to add on Figure 1 that shows the region of interest, a schematic representation of the main circulation features (i.e. Kuroshio, Oyashio currents, + transport of iron from the sea of Okhotsk).

Line 62: It is not clear whether this paper is the one that force NSI-NEM by a 3D circulation fields or whether the novelty lies in the fact that the physics was of better quality than before due to data assimilation. Please clarify.

Line 62: what is the "data assimilated physical field"? Model results with data assimilation. I miss in the introduction a clear description of the objectives of the paper. It is mentioned that the region presents a high variability (and so models have limited capacities in presenting the physical and biogeochemical characteristics of the region) but then the authors focused on monthly averaged values. This is confusing. I also recommend a brief description of the sections of the manuscript.

Line 69 : please clarify which physical fields you are using. I guess that you mean OGCM model outputs.

Line 88: What do you mean by "employed by those"?

Line 113: What about lateral inputs/exports?

Line 114-115: Usually the set of parameters on which the optimization efforts are concentrated are selected based on sensitivity/idenifiability studies. Here this is not really clear how they are selected. (how many parameters are in the model?)

Lines 130-131: Please comment on the assign weights in the cost function. 0.1 $\mu$mol/l seems to be quite low.

Lines 149- +Figure 3: The performances of the model at the two stations with data assimilation are not convincing at all. The model without data assimilation has better performances than the model with data assimilation. The authors mention "The seasonal variations of PS in the Parameter-optimized case (dashed lines) for the two stations simulated by the satellite data (solid lines) were more accurate than those in the Default case (dotted lines)." Looking at Figure 3b, this is not what we can see.

Figure 4: I would recommend to compute errors metrics like bias, RMS, .. in order to quantify more objectively model data misfit and to assess whether the assimilation of data really helps to improve model performances.

Figure 1: please explain why you have different colors for the boxes?

---

## Author Comment (AC1) · 29 Jul 2017

First of all, we should disclose our mistake of drawing Figure 3 in our manuscript (MS).

Referee #1's comment gave us the chance of finding our mistake: As the first step of simulation procedure, using 1-D ecosystem model, we obtained an optimized parameter set for our ecosystem model. It takes 8 model years for a simulated ecosystem to approach an equilibrium state from an initial condition. Our mistake was that Figure 3 was drawn using monthly climatology over the 8 years, although we should have used monthly data only in the last simulated year.

[Figure]

The serious problem was that we described our results in section 3.1 based on the wrong Figure 3 mentioned above, without noticing the error previously. As correctly pointed out by Referees #1 and #2, the optimized case was clearly worse than the default case. Please find the next page that includes the wrong and correct Figure 3. In the correct figure, simulated data in the optimized case (dashed lines) is clearly closer to Satellite data (solid lines) than that in default case (dotted lines).

We made sure that we did use the optimized parameter set obtained for the CORRECT Figure 3 in the submitted manuscript in spite of the wrong Figure 3 in it. Therefore we had NOT found any problem in the following steps of simulation using 3-D model (hence, the rest of manuscript).

We must sincerely apologize to two referees, the topic editor and the people who are interested in this study for your time to read our MS with the wrong Figure 3. We are now rewriting our MS and preparing replies to the referees' comments.
* * *
[Figure]

Figure 3. Seasonal variations of surface phytoplankton biomass in the 1D NSI-MEM and satellite data at (a) St. KNOT and (b) St. S1 are shown as typical observational points of the subarctic and the subtropical regions, respectively.

**Fig. 1.**

---

## Author Response (AR1)

Dear Topical Editor,

Firstly, we sincerely apologize for our mistake of the wrong Figure 3 in the previous manuscript. We thoroughly replaced the previous Figure 3 and edited the section related to the Figure 3 in the revised MS.

Secondly, we thank the Editor, Referee #1 and Referee #2 for their useful comments given to our manuscript, as they very kindly gave us many good suggestions and pointed out our weak parts to be improved.

The referees mainly pointed out that (1) the authors did not obtain reasonable improvement of simulated biomass by the 1-D optimization experiments (e.g., Figure 3 in the previous MS) and (2) there were less evidences in the previous MS to judge the improvement by the parameter optimization.

We responded to the referees' comments point-by-point, as in the pages from the next: "Reply to the referee #1 and Reply to the referee #2". In the revised manuscript (another attached file), our revised manuscript is around 50% rewritten, with red-colored fonts (The main body became from 4,120 words to 5,488 words, and figures became 9 sheets to 12 sheets). In addition, we newly added three figures and descriptions to the revised MS, because of increasing the evidences for the improvement. We hope that this revised manuscript will be found to match acceptable level for your journal.

Finally, we apologize for our delayed submission of the revised manuscript.

Sincerely,
Yasuhiro Hoshiba and co-authors

**Reply to the referee #1:** We express our appreciation to the referee for the careful reading of our paper and giving useful comments. The referee's comments gave us the chance to find that Figure 3 of our discussion paper had a big mistake. We also would like to deeply apologize for taking up your time to read our previous manuscript (MS) with the wrong Figure 3.

Below are our responses to Referee. The referee's comments are in italic style and our corresponded replies are in regular style.

*General comments*

*It is interesting to see how the state-of-art LTL ecosystem model simulates the realistic temporal and spatial evolution of plankton biomass, since it constitutes an important part of the geochemical cycles simulated by a model. The LTL ecosystem model used for the current study is the one including iron cycle and its interaction with biomass distribution, which is considered to be necessary to improve the biomass distribution simulated by a model in high nutrient low chlorophyll (HNLC) regions such as the northwestern Pacific Ocean. I think the basic strategy of the current work is well-considered and reasonable for a step-by-step improvement of the LTL ocean ecosystem model -i.e., tune the model parameters by a 1-D model which is computationally inexpensive, and then apply the tuned parameters for a 3-D ecosystem model and examine how the spatial distribution of plankton biomass is simulated. However, it looks to me that the tactics employed in the study is not necessarily suitable for the purpose of the study as described below. The authors did not obtain reasonable improvement of simulated biomass by the 1-D optimization experiments, probably due to an inappropriate choice of tuning parameters, and failed to simulate realistic biomass distribution in the study area. Since the basic strategy is reasonable, I would recommend the authors to rework this issue addressing the following points.*

I would like to appreciate useful comments by the referee, which gave us the chance for noticing that Figure 3 of our discussion paper was a mistake (Please refer to 'Errata for Figure 3'). According to the reviewer's comments, we revised our MS as detailed below:

*Major points*

*1) I would say that the result obtained from the 1-D parameter optimization is quite miserable and frustrating. If I understand the setup of the experiment correctly, the cost function is composed of only 24 values (12 months _ 2 types of phytoplankton biomass), while the number of optimization parameters is 23. This is mathematically equivalent to optimizing 24 modeled*

*values by 23 model parameters. For such an experiment design, one can expect nearly perfect fit of the modeled values to the observation, if the model appropriately describes the process concerned and the choice of the tuning parameter is appropriate. Nevertheless, the authors failed to fit the modeled phytoplankton biomass to the observed ones, nor even reproduce annual mean phytoplankton biomass: the simulated PS biomass at the subtropical station S1 is nearly 10 times larger than the observed one; the simulated PL biomass at the subpolar station KNOT is more than 5 times larger than the observed one during the winter period (Fig. 3), both of which are essential to describe the dominant species of phytoplankton in each area. To be honest, I do not find any benefit to apply the parameter sets, which provides such a large discrepancy from observation, for 3-D model simulations. I would recommend the authors to redo the optimization experiments by taking the points described below into account.*

We would like deeply to appreciate this comment that gave us the chance of finding our mistake. As you pointed out, in Figure 3, the optimized simulation results were clearly worse than the default results. As explained in the Errata published in the Discussion forum on 29[th] July, 2017, our mistake occurred in the process of data selection for drawing Figure 3. We used the optimized parameter set obtained for the CORRECT Figure 3 in the new MS. As a result, we did not have to change the rest of original results of simulation using 3-D model.

In the correct Figure 3 of the revised MS, simulated data in the optimized case (dashed lines) is clearly closer to satellite data (solid lines) than that in the default case (dotted lines). The cost functions in the optimized case, 1.61 and 0.17 at KNOT and S1, more improved than those in the default case, 13.55 and 1.11, respectively. Therefore, we revised Section 3.1 based on the new Figure 3 in the revised MS, specifically at Line 168-184.

*2)  How did the authors select the parameters used for the optimization? It looks to me that the authors selected a number of physiological parameters for the modeled phytoplankton, while did not select any parameters describing physical processes of the system (e.g., sinking rate, etc). As shown in Fig. 3b, the model exhibits too much PS biomass throughout the year, indicating nutrients in the euphotic zone are repeatedly recycled without being extracted from the system, probably due to insufficient parameterization for sinking/scavenging processes. This situation can be also seen in Fig. 7 - the discrepancy between the modeled and observed NO3 becomes larger after optimization (the unoptimized parameter set gave more reasonable result). Therefore, my recommendation is to reconsider the selection of tuning parameters, so as to tune the nutrient input to/output from the euphotic zone, and examine the improvement of nutrient distribution before examining modeled phytoplankton biomass. Otherwise the derived optimal physiological parameters are not describing the realistic function of phytoplankton physiology.*

The parameters for the optimization were selected, referring to those in Yoshie et al.

(2007). Yoshie et al. (2007) suggested that the selected parameters were influenced more than the other parameters such as sinking rate. The parameter selection was not described in the previous MS, and we added it at Line 132-134 in the revised MS.

It was not enough to describe nutrients-limitation in the previous MS, too. As nitrate is not limiting factor of phytoplankton growth at St. KNOT, vertical distribution of nitrate concentration could not be drastically changed by parameter optimization. We added a discussion of vertical distribution of silicate as a limiting factor, comparing with nitrate in Fig. 10 (b, c) in which vertical distribution of silicate in the optimized case is closer to the observation than that in the default case.

On the other hand, vertical *gradients, not concentrations,* of nitrate and silicate, that are tightly related to the nutrient input/output from the euphotic zone as you pointed out, are much closer than those in the default case in the optimized case. That is, in the optimized case, concentrations of nitrate and silicate both at the depth of around 50 m and 250 m are less than the observed ones, while those at the depth of around 50 m in the default case is higher than those in the optimized case, resulting in much smaller *gradients* in the default case than the observed gradients. As for nutrient concentrations at the depth of 250 m, the parameter optimization cannot improved them, because they are determined by physical processes in the ocean-basin scale as well as local biological processes. Descriptions including the above explanation was added to Line 270-285 of the new MS.

*3)   The current work did not take the advantage of the physical field obtained from the high resolution ocean model with data assimilation. During the last two decades, 3D ocean ecosystem models have been suffered from insufficient descriptions of physical fields obtained from low-resolution ocean models (e.g., reproducibility of mixed layer depth, its spatial distribution and seasonal variation, location of subtropical-subpolar boundary, coastal upwelling, etc.). Although the current study utilizes a physical field which is supposed to be free from such problems, I cannot find any distinguishable improvements compared to the studies using the low-resolution physical field. This is probably due to an insufficient implementation of nutrient cycles in the model, particularly the process describing input/output of nutrients into/from the system. My recommendation is, first of all, optimize the modeled nutrient cycles before tuning, examining or discussing the physiological parameters of the LTL ecosystem model. As far as I know, the coauthors listed here have done a number of excellent works and should have sufficient knowledge and experience for this.*

First of all, I would apologize for our mistake of wrong Figure 3 and for less informative figures, to judge the improvement by the parameter optimization. In order to increase the evidences for the improvement, we newly added horizontal surface correlation maps and vertical distributions of phytoplankton in the meridional section along 165°E to the revised MS as Fig. 5 and Fig. 6, respectively. We also added the vertical temperature and salinity distribution along 165°E for comparing and discussing

the reproducibility of mixed layer depth and location of subtropical-subpolar boundary to the revised MS as Fig. 7. The description as the above was added in Section 3.2 (Line 203-227) of the revised MS.

*Specific comments*

*1)    Line 24-64: Although the authors provided a concise review in introduction, the papers cited here seems to be largely biased toward the papers written by the co-authors of this work. I think it would be one more advantage of this work, if the authors mention the work by other groups.*

  We deleted and added some referenced papers by other groups at Line 38-41 in the revised MS.

*2)    Line 33: vales –> values?*

  Thank you for pointing out.    We corrected it.

*3)    Line 45-49: The construction of the sentence seems strange.*

  We revised the sentence in the revised edition at Line 51-57.

*4)    Line 71-72: The authors described that the physical field used for the 3D experiment is obtained from a 3D-Var data assimilation, in which temperature (T), salinity (S) and sea surface height (SSH) are assimilated. This means increments of T, S and SSH are added to the analysis field in each analysis time step. Is the physical field satisfy the mass conservation after the SSH assimilation? If not, how much amount of artificial sink/source of passive tracers should we expect? Is it not essential for the LTL ecosystem model simulation, particularly close to the sea surface?*

  The physical field used in the offline ecosystem model (NSI-MEM) does not satisfy the mass conservation, but the passive tracers of the NSI-MEM in the offline setting do not have artificial sink/source without other boundary forcing and correction terms.

*5)    Line 76-77: What does "similar" mean? The authors should describe the difference from the cited work.*

We wrote more the detailed description in conjunction with the below 6).

*6)    Line 76-77: How did the authors provide dust flux for dissolved iron? The earth system model (Watanabe et al., 2011) contains iron in the dust? If not, how did the authors define the amount of iron concentration in the dust flux? Description is needed.*

The description of iron process lacked in the previous MS.   We added the description process with some important parameter values to Line 75-79 in the revised edition.

*7)    Line 78: How did the authors define nutrient supply from the river? Does the CORE-2 provide nutrient concentration in river run-off? If not, how the author defined the value?*

The nitrate supply was calculated from the freshwater supply of CORE ver. 2 by the concentration of 29 μmol/l, and the silicate was by 102 μmol/l.   The description was added to Line 79-82 in the revised MS.

8)    *Line 78: How did the authors define the nutrient supply from sediment over the shelf area?*

Nitrate and silicate sources were only from the rivers in this setting. Iron supply was only from the dust origin. The description was added to Line 82 in the revised MS.

*9)    Line 79-80: This sentence needs citation.*

The three-dimensional ecosystem model used in this study has not published yet in any journals, so that we added the web site address as the citation of this model participating to MARine Ecosystem Model Intercomparison Project at Line 86 in the revised MS.

*10)    Line 68-83: Where does the iron in the system come from? Many studies addressed the importance of iron supply from the Sea of Okhotsk to the northwestern Pacific Ocean. How did the authors describe the iron supply from the Sea of Okhotsk?*

  The complete reproduction of the water exchanging between the Sea of Okhotsk and the Pacific is very difficult to model, even with 0.1 degree grids. The iron supply from the Sea of Okhotsk could not be well reproduced in this study.   So, we did not focus on the coastal regions like around the Kuril Islands where the water exchanges between the Pacific Ocean and the Sea of Okhotsk.

*11)    Line 68-78: How did the authors define the initial condition of nutrient distribution (nitrate, silicate and iron)? Description is needed.*

  In the parameter-optimized and SST-dependent cases of the 3-D simulations, the physiological parameters were the same as the default case from 1$^{st}$ Jan. 1985 to 31$^{th}$ Dec. 1996. During the next one year (1997), the simulations were spun-up with the optimized or SST-dependent parameters, which simulation results on 1$^{st}$ Jan. 1998 were used as initial conditions for the 1998-year simulations in this study.   On the other hand, the parameters of the default case were not changed during 1985 to 1998.   The explanation about initial conditions was introduced at Line 104-108 in the revised edition.

*12)    Line 79: Describe the range of restoring boundary layer and the restoring time scale.*

  In order to buffer artificial high concentrations of phytoplankton around the side edge of the model domain, the restoring was conducted only in a few grids of the side edge with the time scale from 43 minutes to 3.6 hours.   The description about boundary conditions was added to Line 82-86 in the revised MS.

*13)    Line 80-82: How about the seasonal cycle of mixed layer depth and its spatial distribution? Is it well reproduced? I'm asking this question because I believe the mixed layer*

*depth is the most important factor to regulate the nutrient supply into the euphotic zone.*

The vertical and horizontal physical field in this region had been confirmed to well reproduce the temperature, salinity and velocity by Usui et al. (2006). Therefore, we think that the mixed layer depth is also reproduced well. According to the comment, we also added the new description and figure (Fig. 7) at Line 223-227 in the revised MS.

*14) Line 82-83: Are the nutrients in an equilibrium state after 1985-1998 integration? Is the nutrient distribution of the equilibrium state consistent with observations (e.g., World Ocean Atlas)?*

The nutrients and phytoplankton distribution do not strictly reach the equilibrium state, due to the sequential change of the physical field. In this study, we focus on the simulation results of surface phytoplankton fluctuation in 1998, and the physiological parameter-estimation for 1998 was conducted. There are not many in situ nutrients data in 1998 in the western North Pacific. Dissolved iron that is the main limitation factor for phytoplankton was hardly observed in 1998. In the revised MS, vertical section of phytoplankton and nutrients distributions along 165°E in June, 1998 from WOA was newly added to Fig. 6.

*15) Line 88-89: The construction of the sentence seems strange.*

We improved the sentence at the revised Line 93-94.

*16) Line 89-90: What does the "similar" mean? The authors should describe which parameter(s) had been changed from Shigemitsu et al. (2012).*

There are some differences of the parameters between 1-D simulation of Shigemitsu et al. (2012) and the 3-D default case in this study. The parameters slightly changed from the 1-D simulation of Shigemitsu et al. (2012) were applied to the 3-D simulation as the default case. The difference from Shigemitsu et al. (2012) was described in the revised MS as new Table 2.

*17)    Line 90: I suggest to use 'control-case' instead of 'default case'.*

  We changed the word in the revised MS, and in the following responses after this 17), we use 'control-case' instead of 'default case'.

*18)    Line 91-97: This experiment design is interesting, while I would suggest the authors to explain the basic philosophy behind this. It looks to me that introducing temperature dependency on many physiological parameters of the LTL ecosystem model is equivalent to rewrite the governing equation drastically, since some of the phytoplankton model parameterization already involve temperature dependency.*

  The SST-dependent case was introduced to reduce an artificial gap in   phytoplankton concentration at the boundary between the two regions (Fig. 1 (b)) due to a sudden change in parameter value.    The description was rewritten at Line 94-97 in the revised MS.

*19)    Line 100-104: Description for the temporal and spatial resolution of the data is needed.*

  In the satellite phytoplankton data, the spatial resolution is approximate 0.042° and the temporal resolution is monthly mean in 1998.    On a daily scale, satellite data have a lot of missing value and is not appropriate for model validation.    The description was added to Line 112-123 in the revised edition.

20)    *Line 105-107: Why the authors used AVHRR data for SST-dependent case, instead of using SST obtained from the physical model? I think this experiment design may introduce a discrepancy: the modeled phytoplankton is controlled by two different temperatures (one is from the physical model and the other is from AVHRR). If I'm wrong, please explain which temperature is used to calculate the modeled phytoplankton biomass.*

  We introduced the SST-dependent case just to smooth parameters around the boundary between two regions used in the Parameter-optimized case.    While we determined

parameter values using annually-averaged SST, there was not a significant difference between the AVHRR data and modeled data.

*21)    Line 114: A description for the selected parameters is necessary. The names of the selected parameters seem to be the same with the definitions in Shigemitsu et al. (2012), while it is not clear to the most of (potential) readers what do they mean. I suggest to implement a short description for each parameter in Table 2.*

  Thank you for the good suggestion.   The description was added to Table 2 of the revised.

*22)    Line 115: Again, what does the "similar" means? The difference should be described.*

  The other non-estimated parameters of Control case were the same as those of Parameter-optimized.    We rewrote the line as follows:
  "The other parameters of the NSI-MEM were the same as those in the Control case." at Line 134-135 in the revised MS.

*23)     Line 119-120: The construction of the sentence is strange, and I cannot really understand the meaning of the sentence. Does the sentence mean "the parameter set which provides the lowest cost is reserved"?*

  The parameter set which provides the lowest cost is reserved, and moreover, the μ-GA applies crossover to other parameter sets which have relatively lower costs for generating new parameter sets.    These process were repeated 2,000 times in this study. The sentence was changed at Line 138-142 in the revised MS.

*24)    Line 125: Why should the number of the population used in the genetic algorithm optimization be the same with the number of tuning parameters?*

  It is known from the previous study (Krishnakumar, 1990) that the number of the population should be similar numbers to that of the estimated parameters from the

perspective of computer resources.

*25)    Line 131: Why do the authors used the same weights for PS and PL? Is it based on uncertainties of satellite-derived biomass?*

It is based on a previous publication by Shigemitsu et al. (2012).   We used the same low value as some weights in the previous study.   We described it at Line 150-151 in the revised MS.

*26)    Line 139: 'too small' –> 'smaller than the prescribed threshold'.*

Thank you for the good comment. We changed the expression.

*27)    Line 118-116: If I understand correctly, the parameter optimization by the 1-D model used a 1-year time window. Why the authors do not use a longer time window for the optimization? If the authors define the cost function by a multi-year window, the cost is more reliable. The computational cost is not essential in this case.*

In this study, we focus on the seasonal variation of phytoplankton.   The physical condition of 1998 was used also in the 1-D model over the 1-year (1998) time window. The scope of this study (i.e. seasonal variation of phytoplankton in 1998) was added at Line 108 in the revised MS.

Considering the referee's comments of the following 28) to 31), and due to the mistake of the previous Fig. 3, the previous Section 3.1 was rewritten entirely.

*28)    Line 150-153: It looks to me that the following two sentences contradict each other; "the PS biomass was larger than the PL biomass at both St. KNOT and St. S1," and "Moreover, diatom, represented as PL, are a major group in the subarctic region.". Isn't it?*

*29)    Line 154-155: How do the authors evaluate the uncertainty of the biomass derived from satellite data?*

*30)    Line 149-159: Why does the optimized case exhibit such a large discrepancy from the satellite data? As I mentioned in the major points, I guess the selected parameters are not relevant to improve the discrepancy.*

*31)    Line 149-159: How much (percentage) is the reduction of the cost compared to the 'default case'? I cannot believe that the 'parameter-optimised case' for S1 gives smaller cost than the 'default case', since the PS biomass exhibits such large discrepancy from the satellite data. Please show the total cost for 'default case' and 'parameter-optimised case', and cost for each month (e.g., a figure with the same abscissa as Fig. 3, while the ordinate is defined by cost for each month).*

*32)    Line 170-172: I'm a bit skeptical to the specification provided here. The authors employed a physical field obtained from an eddy-resolving model, yet they argued that the lack of the small-scale mixing is still responsible for the low-biased biomass close to the cost (or over the shelf). If this is true, what is the advantage to use the eddy resolving physical field in this study? Since Fig. 4 and 9 successfully reproduced an eddying physical field, I guess a lack of nutrient supply from the seabed is a likely reason for the low-biomass close to the cost. How did the authors implement nutrient flux from seabed? Is it suitable to reproduce nutrient cycle over the shelf and/or close to the cost?*

Nutrient flux from the seabed was not considered in this study. As the referee suggested, it might be the reason for the low-biased phytoplankton biomass close to the coast.    However, the main focus of this study is phytoplankton seasonal fluctuation in the pelagic and open ocean (deeper 200 m).    Considering this referee's suggestion, we improved the part (Line 197-202 of the revised MS).

*33)    Line 181-194: I think Fig. 6 (and associated analysis provided here) is an useful measure to characterize the performance of a LTL ecosystem model. But I would say, due to the large discrepancy between observed and simulated biomass (Fig. 3, 4 and 5), the analysis is not necessarily useful. I suggest to redo the analysis after a re-optimization of the model parameters as described in the major points.*

We sincerely apologize for the wrong Figure 3 in the previous MS.    After the

correction to Fig. 3, we think that the estimated parameters and the following analyses are decently useful. In order to verify the simulation validity, horizontal distributions of correlation to satellite data, as well as a comparison with observed data of vertical section along 165°E from WOA, were added as Fig. 5 and Fig. 6 in the revised MS, respectively.

*34) Line 196-203: How did the authors take into account the spatial and temporal representativeness of the modeled phytoplankton (and nutrient) for the comparisons? Since the horizontal distribution of the modeled properties has an eddy-scale fluctuation (e.g., Fig. 4), a direct comparison with in-situ data is meaningful, if and only if the physical model accurately reproduced the location and evolution of respective eddies. I'm not sure this is the case or not, since the physical model assimilated SSH (the reproducibility of realistic eddy fields depends on the spatial and temporal resolution of the assimilated SSH and the assimilation interval). If the location of respective eddies are not necessarily realistic, a mean value should be used for the comparisons (and standard deviation of the field should be used for the measure of uncertainty). Otherwise, we cannot argue which line in Fig 7 is closer to the in-situ data.*

We think that mesoscale eddies are reproduced in some extent in the model, due to the multivariate 3D variational analysis scheme already built in the physical model. As you pointed out, however, there is the possibility that the location of a mesoscale eddy on the station cannot be completely reproduced. We added error bars which depicted the maximum and minimum values in ± 0.3° around the grid of St. KNOT to Figure 10 of the revised edit.

*35) Line 204-218: I'm also skeptical to the usefulness of the analysis and discussions provided here, since the background field for the modeled LTL ecosystem (i.e., nutrient fields) are not thoroughly examined nor confirmed to be realistic. The authors compared the vertical distribution of NO3 between model and observation only at one location. I think it is necessary to check the reality and weakness of the modeled nutrient fields before proceeding analyses for the modeled phytoplankton physiology. I suggest to compare the spatial and vertical profiles of nutrient fields (nitrate, silicate and iron) with available atlas and data sets.*

We newly compared the data in situ and the simulation results of nutrients and phytoplankton concentrations at Line 210-222 in the revised MS.

*36)    I found a number of strange construction of sentences. I think a consultation of the English sentences is necessary.*

We have received English proof-reading before this resubmitting the manuscript.

**Reply to the referee #2:** We express our appreciation to the Referee #2 for giving useful comments for our paper. We also would like to deeply apologize for taking up your time to read the previous manuscript (MS) with the wrong Figure 3. Below are our response to the comments. The referee's comments are in italic style and our corresponded replies are in regular style point-by-point as follows.

*General comments and major points*

*Parameters estimation in biogeochemical models is a critical challenge and assimilation of data is an important tool to better constrain these parameters. This manuscript is thus dealing with a topic of relevance. It is clearly written and presented, figures are of good quality. I have however important criticisms concerning the methodology used in the manuscript. Mainly, I find that the paper does not provide enough convincing arguments that the way data assimilation is used here clearly improves model performances. The use of data assimilation requires a thorough quantitative assessment of model performances before assimilation and after assimilation and this assessment is insufficient here. Only the seasonal cycle of monthly averaged surface concentration of phytoplankton is presented and broadly compared with observation and this is not sufficient. I would recommend that the authors strongly improve that point. Then, I would like that the authors comment of the propagation of errors that may happen when 1) interpolating a global chlorophyll product to their grid (is there not a regional product available for the region?, 2) converting global chlorophyll to PFT, 3) converting chlorophyll to nitrogen values. The paper does not convincingly demonstrate that model performances are really improved thanks to the assimilation of such data (conversely looking at e.g. figure 3, we have the feeling that data assimilation degrades model performances see my details comments below). The readers would be better convinced by the gain obtained by the assimilation of such data if the authors presents a thorough error assessment computing error statistics like bias, RMS,: : : Therefore, I find that the paper cannot be accepted for publication in its present form and would require a substantial revision.*

First of all, I would like to sincerely apologize to you for our mistake of Figure 3 in the previous MS. As you mentioned, the simulation results by the parameter-optimized case were clearly worse than those by the default case in the wrong Figure 3 in the previous MS. We have explained how the mistake occurred in Errata published on the Discussion forum on 29[th] July, 2017. As explained in the Errata, our mistake occurred in the process of data selection for drawing Figure 3. In the new MS, this was corrected. On the other hand, we found that we did not have to change the rest of original results due to the mistake of Figure 3. In the correct Figure 3 of the revised

MS, simulated results in the optimized case (dashed lines) are clearly closer to Satellite data (solid lines) than those in the default case (dotted lines), showing a significant improvement. In addition, we revised descriptions related to the previous wrong Figure 3 (i.e. Section 3.1 of the revised MS).

We agree the referee's suggestions that the previous MS had too less figures and it was difficult to judge the improvement by the parameter optimization. We newly added Figs. 5 and 6 to the revised MS. Figure 5 shows horizontal correlation maps between the model and the satellite, and Fig. 6 depicts vertical distributions of nutrients and phytoplankton. Figure 5 shows a broader horizontal assessment of model results rather than the point comparison (e.g., the previous Fig. 5). Figure 6 is drawn for the assessment of vertical distributions of chlorophyll in the meridional section along 165°E. The results and discussions of the figures were added to Section 3.2 in the revised MS.

As for your comment about interpolating a global chlorophyll product to their grid, the detailed evaluation is difficult, the interpolation might be one of the error sources, as you pointed out. On the other hand, the previous study (Gregg and Casey, 2004) showed that the global satellite chl-a compares well with in situ data even in the WNP region ($r^2$=0.71, RMSE % log error of 31.6). The description was added to Line 120-123 in the revised MS.

As for your comments about an error propagation from chlorophyll to nitrogen values through the conversion from chlorophyll to PFTs, the propagation cannot be assessed at the moment due to a lack of data of the in situ nitrogen-chlorophyll ratio for each PFT matched up with the satellite data (Uncertainties in chlorophyll a and PFTs are published in Gregg and Casey 2004 and in Hirata et al. 2011, respectively). Hence, we used the Redfield Ratio of 6.625 for Carbon-to-Nitrogen ratio, and then the Carbon-to-Chlorophyll values of 50 (PL) and 125 (PS) that result in the N:Chl ratio of 1:1.59 (PL) and 1:0.636 (PS), respectively, as previously used in Shigemitsu et al. (2012). We described this point in the descriptions of Fig. 3 of the revised MS for readers as follows:

"The unit conversion between the simulation data (molN/m$^3$) and the satellite data (gchl-a/m$^3$) is referred to as the nitrogen-chlorophyll ratio of PL= 1: 1.59 and PS= 1: 0.636 (Shigemitsu et al., 2012). The same conversion of nitrogen-chlorophyll is used to Fig. 4, Fig. 6, Fig. 8 and Fig. 10."

*Detailed comments*

*1)   Line 16-17: The authors say "The approach used a one-dimensional emulator that*

*referenced satellite data". Please clarify what you mean by "emulator" and "reference to satellite data". I guess that you mean that the estimation of parameters is based on a 1D simulation using satellite data for constraining biogeochemical parameters.*

We improved the sentence at Line 16-17 of the revised MS.

*2)  Lines 17-18: The comparison with other models that do not assimilate data is only limited to the model used in this paper and cannot be considered as a general feature. Please reformulate.*

The simulation results of NSI-MEM in this study are not compared with those of other models.   We reformulated it at Line 17-19 of the revised MS as follows:
"The 3-D NSI-MEM optimised by the data assimilation improved the timing of a modelled plankton bloom in the subarctic and subtropical regions compared to the model without data assimilation."

*3)  Line 33-35: "Physiological parameters have often been tuned up empirically and arbitrarily, although ecosystem models have recently added more parameters to increase the number of prognostic and diagnostic variables". It is not clear here why there is an opposition of the two parts of the sentence (use of although). The fact that the number of models parameters increases makes it more and more difficult to tune them automatically. On the other, I agree that ecosystem models are more and more complex with the aim to increase their reliability but the increased number of uncontrolled parameters may make them less realistic.*

According to the referee, we rewrote the sentence at Line 43-45 of the revised MS as follows:
"Moreover, physiological parameters have been often tuned up empirically and arbitrarily. The fact that the number of parameters increases with prognostic and diagnostic variables makes it more difficult to tune them."

*4)  Line 36: Please clarify what you mean by ´n a reasonable estimate of the physiological parameters Â˙z Line 40: set and not set?*

This description was rewritten at Line 45-47 of the revised MS as follows:

"In order to reproduce observed data such as spatial distribution of phytoplankton biomass and timing of a plankton bloom, it is required to reasonably estimate the physiological parameters."

*5)    Line 42: I find that the use of "This algorithm"" may be confusing. I guess that here the authors refer to the model NSI-NEM and so I would use "model" instead of "algorithm".*

**"**This algorithm" here means the micro-genetic algorithm (μ-GA). We changed the words to "the μ-GA" at Line 52 in the revised MS.

*6)    Line 45: I guess that the sentence is incomplete. I would add something like 'based on" before following the…*

We improved the sentence at Line 50-57 of the revised MS.

*7)    Lines 44-49: This part would be better placed in the section on materials and methods*

We would like to put this part in Introduction, because the point here is that it was developed by the previous study in Shigemitsu et al. (2012).  We revised the part at Line 50-57 in the revised MS.

8)    *Line 50 : "and the Parametres-optimized approach" this is vague I would say and a micro genetic algorithm to optimize..*

The part was eliminated, due to the substantial revision of Introduction.

*9)    Lines 55-60: I would suggest to add on Figure 1 that shows the region of interest, a schematic representation of the main circulation features (i.e. Kuroshio, Oyashio currents, + transport of iron from the sea of Okhotsk).*

According to the referee's suggestion, the schematic representation of the main circulation features were added to Figure 1 (a) in the revised MS.

*10)    Line 62: It is not clear whether this paper is the one that force NSI-NEM by a 3D circulation fields or whether the novelty lies in the fact that the physics was of better quality than before due to data assimilation. Please clarify.*

In this study, while the NSI-MEM runs offline using the physical fields simulated (and assimilated) using an OGCM, the physiological parameter estimation (using biological data assimilation) was also applied simultaneously to the NSI-MEM using µ-GA scheme .   We clarified the part in conjunction with the below 11).

*11)    Line 62: what is the "data assimilated physical field"? Model results with data assimilation. I miss in the introduction a clear description of the objectives of the paper. It is mentioned that the region presents a high variability (and so models have limited capacities in presenting the physical and biogeochemical characteristics of the region) but then the authors focused on monthly averaged values. This is confusing. I also recommend a brief description of the sections of the manuscript.*

The physical field were also data assimilated by 3D-VAR, but that was not conducted by our study. For understandability and comprehensibility, we improved the part and added the brief description of the sections at Line 58-63 in the revised MS.

*12)    Line 69: please clarify which physical fields you are using. I guess that you mean OGCM model outputs.*

We added some words for clarity at Line 66-71 of the revised MS.

*13)    Line 88: What do you mean by "employed by those"?*

We meant "For each province, the respective parameters estimated by the µ-GA and the 1D NSI-MEM were employed to those in the 3D NSI-MEM.".   The sentence was

rewritten at Line 93-94 in the revised MS.

14) *Line 113: What about lateral inputs/exports?*

 Lateral inputs/exports, horizontal advection and diffusion were not considered in the 1D model. The description relating to that was added to Line 263-265 in the revised MS as follows:
 "These suggests that effects of horizontal advection such as mesoscale eddy is important for the daily reconstruction of the profile in the upper layer as the effects are not included in the 1D model."

15) *Line 114-115: Usually the set of parameters on which the optimization efforts are concentrated are selected based on sensitivity/idenifiability studies. Here this is not really clear how they are selected. (how many parameters are in the model?)*

 We determined that the 23 of 107 physiological parameters used in NSI-MEM were optimized, according to the previous study (Yoshie et al., 2007). The description was added to Line 131-134 in the revised MS.

16) *Lines 130-131: Please comment on the assign weights in the cost function. 0.1 μmol/l seems to be quite low.*

 We adopted the same low value as some weights used in Shigemitsu et al. (2012) who successfully implemented the μ-GA for our target region. We describe it on Line 150-151 in the revised edition.

17) *Lines 149- +Figure 3: The performances of the model at the two stations with data assimilation are not convincing at all. The model without data assimilation has better performances than the model with data assimilation. The authors mention "The seasonal variations of PS in the Parameter-optimized case (dashed lines) for the two stations simulated by the satellite data (solid lines) were more accurate than those in the Default case (dotted lines)." Looking at Figure 3b, this is not what we can see.*

We sincerely apologize for our mistake of the previous Figure 3.   We totally changed Section 3.1 related to Figure 3 in the revised MS.

*18)    Figure 4: I would recommend to compute errors metrics like bias, RMS, .. in order to quantify more objectively model data misfit and to assess whether the assimilation of data really helps to improve model performances.*

Thank you for good suggestion.   We newly added Figure 5 and Figure 6 to the revised MS, to increase the evidence for improvement by the parameter optimization.

*19)    Figure 1: please explain why you have different colors for the boxes?*

The colors are divided by the features (e.g., phytoplankton, zooplankton, particulate matter and nutrient).   The description was added to the caption of Figure 2 in the revised MS.

---

## Author Response (AR2)

Dear Topical Editor,

We would like to thank the Editor, Referee #1 and Referee #2 for handling and reviewing our manuscript.

We responded to the referees' comments point-by-point in "Reply to the referee #1 and Reply to the referee #2". In the revised manuscript, we improved the figures and descriptions. We hope that this revision will be found acceptable for your journal.

Sincerely,
Yasuhiro Hoshiba and co-authors

**Reply to the referee #1:** We express our appreciation to the referee for the careful reading of, and the comments to, our paper. The below are our responses to the comments. The referee's comments are expressed in italic style and our corresponded replies are in regular style.

*Having look the revised Fig. 3 and associated results, I found the parameter estimation by genetic algorithm gives a reasonable result and the manuscript is now ready for scientific discussion and review. As a whole, the manuscript was substantially improved in comparison to the one for the first submission, while there are some important points, for which more explanations/clarifications are necessary; many figures need to be improved; some part of the text should be rewritten for readability. For these reasons another round of revision is necessary.*

We would like to appreciate your time to review our previous MS. According to your comments, we further revised our manuscript (MS) as detailed below:

*Specific comments*
*1)    Introduction gives concise and good summary for previous work, and points out the necessity of the current work.*

Thank you.

*2)    line 56. ".. classical Michaelis-Menten equation .." needs reference.*

The reference was added in Line 56-57 in the revised MS.

*3)    line 69-71. Again I have a question/concern about the use of physical field obtained from data assimilation, particularly since a 3D-Var system is used in this study. In reply to referee #1, the authors wrote "The physical field used in the offline ecosystem model (NSI-MEM) does not satisfy the mass conservation, but the passive tracers of the NSI-MEM in the offline setting do not have artificial sink/source without other boundary forcing and correction terms". I don't really understand how the authors achieve such a set-up, since the conservation of passive tracers depends on the conservation of the physical field (i.e., volume). The passive tracers in LTL models are generally represented by concentration in a grid cell. If there is an increment of SSH by assimilation, the volume of the corresponding cell changes, which automatically leads to change*

*of passive tracer amount (if you don't change the concentration). How did the authors handle this issue? If the authors implemented a scheme which preserves amount of passive tracers even with the volume change, then the concentration of passive tracers undergoes artificial change. How much impact do you see in this case? I'm asking this question because the observed and modeled vertical section of T (Fig. 7) exhibits difference around the inter-gyre boundary (approx. 40 degree N), implying not a small amount of SSH increment might be added in this frontal zone. I think this is an important point to be checked and discussed somewhere in the manuscript before going LTL model analyses, since a use of assimilated physical field for ecosystem modeling is (probably) one of the way to go in the future.*

We meant that temperature and salinity were not conserved by the 3D-Var system, but SSH and ecosystem tracers was actually conserved. Because the 3D-Var system used in this study only changes the temperature and salinity, and does not directly change the SSH and velocity (Fujii and Kamachi, 2003). Therefore, the amount of water mass is also conserved. We added the information at Line 72-74 in the revised MS.

*4)     line 82. The authors describe "iron supply was only from the dust in the model setting", while in introduction they pointed out that "The source of iron for the WNP region is not only from air-born dust but also from iron transported in the intermediate water from the Sea of Okhotsk to the Oyashio region" (line 31-33), which seems to me contradicting. A justification for the model setup or discussion on the effect of missing iron source is necessary.*

Our model is not perfect and has caveats. We added a discussion on the effect of missing iron source to Line 295-298 in the revised MS.

*5)     line 95-102. The authors conduct a model run with SST-dependent physiological parameters, while at the same time they also stated physiological parameters (may) change with other conditions e.g., nutrient abundance, light intensity (line 303- ). An explanation is necessary here, why the authors chose SST field to smooth the transition between the two assimilated stations.*

The sentence (L303) quoted by the Referee was described in the context of phytoplankton physiology. On the other hand, physiological parameters estimated using the SST gradient include those of not only phytoplankton but also zooplankton. We chose SST because it directly affects physiological parameters of both phytoplankton and zooplankton whereas nutrients and light are essentially related to phytoplankton only. We added this philosophy to Line 101-104 in the revised one.

*6)     line 107. "The parameters values" --> "The parameter values".*

  Thank you. It was amended.

*7)     line 107-108. Why the authors selected the specific year 1998? I asked this question in my first review but the authors did not provide reasonable answer. I'm asking this question because a parameter estimation by multi-year condition gives more reliable result. If you use the data from short period, the estimated parameters may be deformed so as to make best much for specific condition. In such a case you lose generality of the estimated parameters and difficult to apply them for interpretation of plankton physiology.*

  Limited computer resource was the largest reason why we chose the single year (1998); the 3D and μ-GA simulations are computationally costly (e.g. the μ-GA approach requires 2,000 "generations" to converge parameter values). We partly agree with Referee in that the generality of the estimated parameters may be increased when multi-year data is used for the parameter optimization, although such a treatment may require acclimation/adoption of organism to be well explained in the model. We incorporated this discussion in Line 311-319 in the revised MS.

*8)     line 167-184. Now I found the result is reasonable and the GA optimization works well.*

  Thank you.

*9)     line 203-209. What can we learn from this analysis? It looks to me just giving a duplicated information with Fig. 9. If the authors intend to keep this part, more explanation is needed for the purpose and necessity of this analysis.*

  In response to a comment from the other referee, Figure 5 was added, to show a broader horizontal assessment of our results in addition to the point assessments at KNOT and S1 stations shown in Figure 9. We revised some sentences at Line 235-237 of the revised MS.

*10)     line 230. The construction of the sentence seems strange.*

  The sentence was changed together with 9) above.

*11)	line 233. "maximum" --> "the maximum".*

The word was eliminated, due to the revision below.

*12)	line 229-249. Some sentences are tedious and preventing easy-reading. For instance, line 233-236 "At St. S1, the timing of (the) maximum phytoplankton ... " can be shorten as "At St. S1, OPT case reasonably captures the timing of the phytoplankton bloom, although the amplitude is slightly overestimated." Readers already know that you compare CTRL and OPT in this section, therefore you don't have to repeat "compared to CTRL case" many times. In relation to the above comment, I suggest to use short abbreviations, e.g., control case --> CTRL, parameter-optimised case --> OPT, SST-dependent case --> T-OPT etc. for easy-reading.*

We simplified the sentences in the revised MS.

*13)	line 263-265. This is one of the interesting results of this study (from my point of view), since you quantitatively showed how much difference occurs on the simulated biomass due to effect of horizontal advection, using 1D and 3D models which have exactly the same LTL model function. I suggest to briefly mention this in conclusion.*

We briefly mentioned the difference on Line 327-328 in the revised MS.

*14)	line 263 and error bars in Fig. 8 and 10. Why the authors employ 0.3 degree range to define the uncertainty? Is this an autocorrelation scale of observed chlorophyll (Fig. 8) or ocean structure (Fig. 10)? An explanation or justification is needed (you can easily find such scale estimates in literature).*

It is not an autocorrelation scale. Effects of advection by a mesoscale eddy would be expected within the range of ±0.3 degree (about six grids) in the physical field, because the radius scale and the lifetime of a general mesoscale eddy are O(100 km) and ⩾16 weeks, respectively (Chelton et al., 2011). We added the explanation at Line 268-270 in the revised MS.

*15)	line 286-287. I don't understand the meaning of this sentence. What is the verb of this sentence?*

The sentence was revised (Line 294 in the new version).

*16)     line 322-325. I am skeptical to the statement in this paragraph. If we see the prescribed range (i.e., min and max) and estimated values of parameters in Table 2, many parameters go to its upper or lower prescribed bounds, indicating the optimization result (i.e., optimized parameter set) is strongly constrained by the prescribed bounds. In other words, the GA optimization did not search the entire parameter space freely due to the bounds. This means that the consistency between the physiological parameters and those obtained from in-situ observation (line 323) is not necessarily guaranteed by the current experiment setup (the consistency is already imposed in the experiment design). If the authors really intend to confirm the consistency, the prescribed ranges should be widen beyond the current ranges, and see whether the parameters still stay within the range of in-situ estimated values. From this point of view, I suggest to revise the concluding sentence.*

  Thank you for the useful comments. As suggested, we revised the concluding sentence.

*17)     The quality of the figures is very poor and is not suitable for publication (except Fig. 2). They should be totally redrawn.*

The figures were redrawn.

*-- Fig. 3. Use a log-scale for the ordinate, otherwise it is difficult to distinguish PL lines in panel (b).*

  We have tried it before, but the variations of PS lines became hard to read. As PL in the panel (b) is close to zero (i.e. PL is almost absent), we decided to keep the ordinate in linear scale. We added the information (i.e., PL is almost zero) at Line 190-191 in the revised MS.

*-- Fig. 6, 7, 10 and 11. Use the same vertical range for consistency*

*-- Fig. 3, 8, 10, 11. Apply a consistent manner for color and line type in these figures, e.g., dotted-line for 1D case, solid-line for 3D case; red line for CTRL, blue line for OPT, green line for T-OPT, black line for OBS. etc. The different rules between different figures makes readers confused.*

*-- Fig 8 and 10. Use color shade (with transparency) instead of the error bars. In some cases the error bars are stacked each other and not distinguishable.*

*-- Fig 11. "(/day)" --> "[day^{-1}]".*

*-- Fig. 3, 5, 6, 7, 8, 10, 12. Divisions and labels for abscissa, ordinate and color bars should be reconsidered for easy-reading.*

*-- Fig. 3, 8, 9, 10, 11. Put appropriate legend for line type and color with examples, in empty space in panels.*

*-- Fig. 1. Use the same panel size for (a), (b) and (c), since the practical information in each panel is nearly the same.*

**Reply to the referee #2:** We express our appreciation to Referee #2 for constructive and positive comments to our manuscript. Below are our response to the comments. The original referee's comments are written in italic style and our corresponding replies are in regular style.

*I have read the revised version of the Hoshiba et al manuscript as well as their answers to my criticisms. The authors significantly improve the manuscript: they add some text that clarify a lot of issues, produce new figures in order to reinforce the performances of the model with parameters estimations and most importantly provide a new version of Figure 3.*
*Even if I am still not fully convinced by the significance of the improvement of the data assimilation experiment on the quality of 3D model results (I would have liked to see more error skills like bias, RMS), I think that the authors did a lot of efforts for improving their work and answering our criticisms. I still have some comments.*

We would like to appreciate your understanding of revision of our manuscript (MS).

*Specific comments*
*1)     Line 220: what do you mean by effective nutrient? Limiting nutrient?*

Yes, we meant limiting nutrient. We clarified it in the revised MS.

*2)     Line 271: please clarify what is the mean iron growth rate and give the units. Iron uptake rate? So, the production increases because you change the parameters describing the iron limitation which results that your phytoplankton is not anymore limted by iron. How realistic are your calibrated parameters? (I am not an iron expert and so I cannot realize if the iron uptake efficiency is overestimated or not)*
*Please clarify how the parameters are set at the interface between the "KNOT" and S1 regions? Are you imposing a sharp gradient or are you using some smoothing? From the figures it seems that it is the first option and this results in sharp transition of the simulated biogeochemical fields (e.g. figure 6c). You mention in section 3.5 that the version of the parameters that is smoothed with SST does gives worse results compared to the other one. DO you know why?*

In the present context, the word "the mean iron growth rate" was used to express an average value of the growth rate limited by iron. The limited growth rate (molN/m$^3$/day)

was standardized for phytoplankton group (i.e. it was mathematically divided by PS or PL biomass ($molN/m^3$)), hence its unit is day$^{-1}$. The limited growth rate is a limitation function of growth rate by either nitrogen, silicate or dissolved iron. The smallest rate among them indicates the largest limitation to the rate of phytoplankton's photosynthesis. We added the unit and the information at Line 275-281 in the revised MS.

It is difficult for us, all modelers to know how realistic the parameters used in ecosystem models are. So, we conducted the parameter estimation. We improved the expression about the future task of the estimated parameters in Conclusions in revised MS.

We imposed a sharp gap between the subarctic (KNOT) and subtropical (S1) regions in Parameter-optimised case, and a smoothed boundary in the SST-dependent case. In the performance, the SST-dependent case was slightly worse than the parameter-optimised at the grids of KNOT and S1, potentially due to advection and diffusion from the grids surrounding. However, the SST-dependent method is useful for eliminating the boundary gap and improving performances in some regions, as in Section 3.5.

*3)    Figure 10: 1D model results are not represented here in Figure 10 b and 10c.*

We show 1D model results only in Figure 10 (a), because our main focus is phytoplankton's biomass and variation. We revised the caption of Figure 10.

*4)    Figure 11: Please clarify what is represented in this Figure: maximum growth rate \* limitation function by either nitrogen, silicate and dissolved iron? Uptake rate of nitrogen, iron?*

These represent the growth rates limited by nitrogen, silicate and iron. The smallest rate among the three rates most limits phytoplankton's photosynthesis. The growth rate limited by iron is smallest in this situation, therefore it dominates the rate of photosynthesis. We added the information to the caption of Figure 11 and at Line 275-281 in the revised MS.

---

## Author Response (AR3)

**Reply to the referee #1:** We express our appreciation to the referee for the careful reading of our paper and giving comments. The below are our responses to Referee. The referee's comments are in italic style and our corresponded replies are in regular style.

*As a whole the manuscript becomes better than the previous version, while I think it still needs a further clarification at least the points mentioned below. In addition, I'm still frustrated to the discussion, for example, those in lines 311-319, which looks to me contradicting (or inconsistent) to the aim of this study described in introduction. Anyway, it might be a matter of taste, and I have no further comments contributing to improvement of the manuscript.*

I would like to appreciate your understanding of our previous revision. According to the referee's comments, we revised our MS (manuscript) as detailed below:

*Specific comments*
*1)     Line 70-74: The description of the dynamical field used for the experiment is still ambiguous (or confusing). Additional sentence(s) are necessary for clarification. The manuscript describes that "... 3D variational analysis scheme that synthesizes the observed information such as temperature, salinity and sea surface height", while it also describes "... but the amount of water mass was conserved (Fujii and Kamachi, 2003)", without any reasoning. The following sentence in the response from the author also made me confusing: "Because the 3D-Var system used in this study only changes the temperature and salinity, and does not directly change the SSH and velocity". What does "directly" mean? Are SSH/velocity changed "indirectly"? The answer is "No", if I understand correctly the 3DVar system in Fujii and Kamachi (2003). I suggest to revise line 72-74 as follows,*
*"The T/S fields are obtained from a free model run plus T/S increments. These increments are derived from the 3DVar system so as to minimize model - observation misfits of T and SSH. Different from the T/S fields, dynamical fields (e.g., u, v, psi, SSH etc.) are not modified by the data assimilation (i.e., the physical field holds mass conservation, which is necessary to run the ecosystem model with a consistent manner)."*
*If I'm wrong and the authors disagree the revision, further explanation is necessary in this part.*

We addressed the more information about the 3D variational analysis scheme at Line 72-75 in the MS.

*2) Line 210-216: Since the time series are composed of monthly mean of 12 data, the 2-month lag-correlation is calculated by only 10 data points, which looks to me not always sufficient for scientific robustness. I suggest to provide a limitation regarding robustness of the result due to the short time series.*

We added the maps of significance level in Figure 5 and mention it at Line 212-215 in MS.

*3) line 243-254: I agree that the diagram shown in Fig. 9 is a concise way to summarize the results, whereas it is misleading to show this figure without mentioning a limitation of this analysis. Since the current experiment deals with monthly mean data, the phase (angle of the arrows) have a resolution of 30 degree. Therefore "True" does not mean a perfect match to the satellite data, while it means a match with 1-month (30 degree) error range.*

We deleted "perfect" and revised the expression at Line 249 and the figure caption in the MS.

*4) Line 290: ".. in which much smaller gradients than the observed gradients are found". This is valid for silicate only. Isn't it?*

We revised the sentence at Line 293 in the MS. We meant that the vertical gradients both of silicate and nitrate in the OPT are closer to the observed data than those in the CTRL.

*5) line 311-319: After reading the paragraph here, I'm wondering what is the benefit of the current parameter optimization. If the ecosystem model parameters are changing with the condition mentioned in the manuscript (e.g., SST, nutrient abundance, light intensity, ..), what can we learn about the mechanisms governing ecosystem behavior from the optimized parameter sets for a specific situation? It looks to me that the dependence of the parameters on such environmental condition reveals insufficiency of the model formulation. Maybe further discussion/explanation which is consistent with the aim of this study should be necessary here. I guess at least the last co-author of the manuscript has an idea explaining the background philosophy of this experiment.*

As the referee suggested, our philosophy for data assimilation and ecosystem models was described at Line 319-326 in MS.